# Blood metabolomic profiling reveals new targets in the management of psychological symptoms associated with severe alcohol use disorder

Sophie Leclercq[1†], Hany Ahmed[2†], Camille Amadieu[3], Géraldine Petit[4], Ville Koistinen[2,5], Quentin Leyrolle[3,6], Marie Poncin[4], Peter Stärkel[7], Eloise Kok[8,9], Pekka J Karhunen[9], Philippe de Timary[4], Sophie Laye[6], Audrey M Neyrinck[3], Olli K Kärkkäinen[10], Kati Hanhineva[2,5‡], Nathalie Delzenne[3*‡]

[1]Laboratory of Nutritional Psychiatry, Institute of Neuroscience, UCLouvain, Université catholique de Louvain, Brussels, Belgium; [2]Food Sciences Unit, Department of Life Technologies, University of Turku, Turku, Finland; [3]Metabolism and Nutrition Research Group, Louvain Drug Research Institute, UCLouvain, Université catholique de Louvain, Brussels, Belgium; [4]Department of Adult Psychiatry, Cliniques Universitaires Saint-Luc and Institute of Neuroscience, UCLouvain, Université catholique de Louvain, Brussels, Belgium; [5]School of Medicine, Institute of Public Health and Clinical Nutrition, University of Eastern Finland, Kuopio, Finland; [6]Université de Bordeaux, INRAE, Bordeaux INP, NutriNeurO, UMR 1286, Bordeaux, France; [7]Department of gastro-enterology, Cliniques Universitaires Saint Luc, Brussels, Belgium; [8]Department of Pathology, University of Helsinki, Helsinki, Finland; [9]Faculty of Medicine and Health Technology, Tampere University and Fimlab Laboratories, Tampere, Finland; [10]School of Pharmacy, University of Eastern Finland, Kuopio, Finland

*For correspondence:
nathalie.delzenne@uclouvain.be

†These authors contributed equally to this work
‡These authors also contributed equally to this work

## eLife assessment

This study provides **valuable** insights and allows for hypothesis generation around diet-microbe-host interactions in alcohol use disorder. The strength of the evidence is **convincing**: the work is done in a rigorous manner in a well-described cohort of patients with AUD before and after withdrawal. There are several weaknesses, including validating the metabolites identified by metabolomics, the cross-sectional study design, the lack of a healthy control group, and the descriptive nature of such clinical cohort studies. Nevertheless, the study provides a wealth of new data that may be the basis for future studies that test causality and elucidate the role of single metabolites in the psychiatric sequela of AUD.

## Abstract

**Background:** Alcohol use disorder (AUD) is a global health problem with limited therapeutic options. The biochemical mechanisms that lead to this disorder are not yet fully understood, and in this respect, metabolomics represents a promising approach to decipher metabolic events related to AUD. The plasma metabolome contains a plethora of bioactive molecules that reflects the functional changes in host metabolism but also the impact of the gut microbiome and nutritional habits.

**Methods:** In this study, we investigated the impact of severe AUD (sAUD), and of a 3-week period of alcohol abstinence, on the blood metabolome (non-targeted LC-MS metabolomics analysis) in 96 sAUD patients hospitalized for alcohol withdrawal.

**Results:** We found that the plasma levels of different lipids ((lyso)phosphatidylcholines, long-chain fatty acids), short-chain fatty acids (i.e. 3-hydroxyvaleric acid) and bile acids were altered in sAUD patients. In addition, several microbial metabolites, including indole-3-propionic acid, p-cresol sulfate, hippuric acid, pyrocatechol sulfate, and metabolites belonging to xanthine class (paraxanthine, theobromine and theophylline) were sensitive to alcohol exposure and alcohol withdrawal. 3-Hydroxyvaleric acid, caffeine metabolites (theobromine, paraxanthine, and theophylline) and microbial metabolites (hippuric acid and pyrocatechol sulfate) were correlated with anxiety, depression and alcohol craving. Metabolomics analysis in postmortem samples of frontal cortex and cerebrospinal fluid of those consuming a high level of alcohol revealed that those metabolites can be found also in brain tissue.

**Conclusions:** Our data allow the identification of neuroactive metabolites, from interactions between food components and microbiota, which may represent new targets arising in the management of neuropsychiatric diseases such as sAUD.

**Funding:** Gut2Behave project was initiated from ERA-NET NEURON network (Joint Transnational Call 2019) and was financed by Academy of Finland, French National Research Agency (ANR-19-NEUR-0003-03) and the Fonds de la Recherche Scientifique (FRS-FNRS; PINT-MULTI R.8013.19, Belgium). Metabolomics analysis of the TSDS samples was supported by grant from the Finnish Foundation for Alcohol Studies.

## Introduction

Alcohol use disorder (AUD) is a global health problem accounting for substantial difficulties for the individuals who consume, their related persons, and for the society. The biochemical mechanisms that lead to this disorder are not yet fully understood, and in this respect, metabolomics represents an interesting approach to decipher metabolic events related to AUD. Improving our understanding of the pathology could lead to discovering potential novel targets for therapies (*Voutilainen and Kärkkäinen, 2019*).

Alcohol consumption clearly leads to alterations of the circulating metabolome (*Voutilainen and Kärkkäinen, 2019*; *Jaremek et al., 2013*; *Würtz et al., 2016*; *Lehikoinen et al., 2018*). For instance, changes in the levels of lipids (fatty acids, phosphatidylcholine, steroids) and amino acids (glutamine, tyrosine, alanine, serotonin, asparagine) are commonly observed and, interestingly, some changes in the blood metabolite profile precedes the emergence of alcohol use related diseases, such as lower levels of serotonin and asparagine (*Kärkkäinen et al., 2020*). Metabolomics studies conducted in rodent models of alcohol exposure have mostly targeted the liver tissue or urine (*Gao et al., 2011*; *Ma et al., 2020*; *Fernando et al., 2010*). In human, plasma or serum metabolomics studies have tested the effects of low/moderate/excessive alcohol intake (*Zhu et al., 2021*) but are rarely performed in clinical populations of severe AUD (sAUD) patients, including analysis of CNS tissues (*Kärkkäinen et al., 2021*). Furthermore, the methodology could differ across studies, some using nuclear magnetic resonance (NMR) while others using more sensitive mass spectrometry (MS) coupled with liquid or gas chromatography (LC or GC), leading to the detection of different metabolites.

The human blood metabolome consists of (1) small molecules that directly represent the functional changes in host metabolism, (2) metabolites produced by intestinal micro-organisms and (3) metabolites originating from nutrition of other exogenous sources like drugs (*Voutilainen and Kärkkäinen, 2019*; *Bar et al., 2020*).

The aims of this study are multiple. First, we investigated the impact of sAUD on the blood metabolome by non-targeted LC-MS metabolomics analysis. Second, we investigated the impact of a short-term alcohol abstinence on the blood metabolome followed by assessing the correlations between the blood metabolome and psychological symptoms developed in sAUD patients. Last, we hypothesized that metabolites significantly correlated with depression, anxiety or alcohol craving could potentially have neuroactive properties, and therefore the presence of those neuroactive metabolites was confirmed in the CNS using post-mortem analysis of frontal cortex and cerebrospinal fluid of persons with a history of heavy alcohol use.

Our data bring new insights on xenobiotics- or microbial-derived neuroactive metabolites, which can represent an interesting strategy to prevent or treat psychiatric disorders such as sAUD.

# Materials and methods

## Study design and participants

A total of 96 sAUD patients hospitalized for a 3-week detoxification program in the alcohol withdrawal unit at Cliniques Universitaires Saint-Luc, Brussels, Belgium were recruited. These patients belong to two different cohorts, namely Alcoholbis (patients recruited in 2015 and 2019) and GUT2BRAIN (patients recruited in 2018–2019; Table 1). The severity of AUD was evaluated by a psychiatrist using the Diagnostic and Statistical Manual of Mental Disorders (DSM) criteria, fourth edition (DSM-IV; Alcoholbis cohort) or fifth edition (DSM-5; GUT2BRAIN cohort). Patients evaluated with the DSM-IV received the diagnosis of 'alcohol dependence', while the patients evaluated with the DSM-5 received the diagnosis of 'severe alcohol use disorder' (six or more criteria). To simplify, we used the term 'sAUD' (for severe alcohol use disorder) that includes both diagnosis (sAUD and alcohol dependence). Patients were eligible if they had been drinking until the day of admission to the detoxification unit or the day before, and if they also did not suffer from inflammatory bowel disease, other chronic inflammatory diseases (such as rheumatoid arthritis) or cancer, nor from metabolic disorders such as obesity (BMI >30 kg/m$^2$), diabetes and bariatric surgery, or severe cognitive impairment (MMSE <24). We also excluded subjects who had taken antibiotics, probiotics, or prebiotics in the 2 months prior to enrolment and those who were taking non-steroidal anti-inflammatory drugs or glucocorticoids within 1 month of inclusion. Patients with known cirrhosis or significant liver fibrosis (≥F2) detected by Fibroscan (>7.6 kPa) on the day of admission were also excluded from the study. No other psychiatric diseases, and no other addiction (except tobacco) have been diagnosed in these patients.

sAUD patients were tested twice, on the day following their admission (T1) and on days 18–19 (T2) corresponding to the last days of the detoxification program. The patients of the GUT2BRAIN cohort

**Table 1.** Clinical features of the study participants.

|  | Alcoholbis cohort | GUT2BRAIN cohort | All | p-value |
|---|---|---|---|---|
| Number of subjects | 48 | 48 | 96 |  |
| Age | 46±10 | 48±9 | 47±10 | 0.24 |
| Gender<br>Men, n (%)<br>Women, n (%) | 34 (71%)<br>14 (29%) | 30 (62.5%)<br>18 (37.5%) | 64 (67%)<br>32 (33%) | 0.39* |
| Smoking status<br>Active smoker (%)<br>Non-smoker (%) | 77<br>23 | 79<br>21 | 78<br>22 | 0.83* |
| Alcohol history<br>Alcohol consumption (g/day)<br>Duration of drinking habits (years)<br>Number of withdrawal cures<br>AUDIT score<br>DSM5 score | 151±112<br>17±10<br>1±2<br>31±7 | 139±73<br>16±11<br>2±2<br><br>8±2 | 145±94<br>16±11<br>1.8±2.3 | 0.54<br>0.76<br>0.015† |
| Depression<br>Score at T1 | 23±11 | 26±12 | 25±12 | 0.26 |
| Anxiety<br>Score at T1 | 44±11 | 46±15 | 45±13 | 0.27 |
| Alcohol craving<br>Total score at T1<br>Obsession score at T1<br>Compulsion score at T1 | 20±7<br>9±5<br>11±3 | 25±6<br>11±4<br>14±3 | 22±7<br>10±4<br>13±3 | 0.00†<br>0.26<br>0.00† |

Results are means ± standard deviations. Independent t-tests to compare ALCOHOLBIS *versus* GUT2BRAIN cohorts.

*chi-square test for categorical variables.

†p<0.05.

were initially enrolled in a randomized, double-blind, placebo-controlled study assessing the impact of prebiotic fiber supplementation on the gut-liver-brain axis (*Amadieu et al., 2022a*; *Amadieu et al., 2022b*). For this reason, only biological and psychological data obtained at admission (T1), and before the beginning of the prebiotic/placebo treatment, were considered. The patients of the ALCOHOLBIS cohort did not take part in any other clinical study during the 3-week hospitalization stay. Therefore, for this cohort, data obtained at both times of alcohol withdrawal (T1 and T2) were considered. Thirty-two healthy controls (13 from the GUT2BRAIN cohort and 19 from the ALCOHOLBIS cohort) matched for age, gender and BMI with no AUD (Alcohol use disorders test [AUDIT] score <8 in males and <7 in females) were also recruited using flyers posted in Brussel's public setting (Table S1 in *Supplementary file 1*). The inclusion/exclusion criteria were the same as for sAUD patients except for the alcohol-related items. Healthy controls and sAUD patients were not matched for smoking status.

The study was approved by the 'Comité d'éthique Hospitalo-facultaire Saint-Luc UCLouvain' (2017/04JUL/354 and 2014/31dec/614, identification number NCT03803709 at ClinicalTrials.gov). The study has been carried out in accordance with The Code of Ethics of the World Medical Association and followed the ethical guidelines set out in the Declaration of Helsinki. All participants provided written informed consent in compliance with the European law 2001/20/CE guidelines.

For investigating presence of the potentially neuroactive metabolites in the CNS, we used metabolomics data from frontal cortex (Broadman area 9) and CSF samples from the Tampere Sudden Death Study (TSDS) cohort, which have been described in detail elsewhere (*Kärkkäinen et al., 2021*). TSDS was collected from forensic autopsies done in the area of the Pirkanmaa Hospital District during 2010–2015, a total of 700 subjects. Out of these we identified 97 heavy alcohol users based on autopsy reports and medical records (diagnosis of alcohol-related diseases: ICD-10 codes F10.X, G31.2, G62.1, G72.1, I42.6, K70.0-K70.4, K70.9, and K86.0, or signs of heavy alcohol use in the clinical or laboratory findings, e.g., increased levels of gamma-glutamyl transferase, mean corpuscular volume, carbohydrate-deficient transferrin). Lack of these findings was inclusion criteria for the control group (n=100), most of whom had died due to cardiovascular diseases (Table S2 in *Supplementary file 1*). Samples were stored at –80 °C until use.

## Assessment of psychological symptoms

sAUD patients were tested for depression, anxiety and alcohol craving with self-reported questionnaires: the Beck Depression Inventory [BDI] (*Beck et al., 1996*), the State-Trait Anxiety Inventory (STAI form YA)(*Spielberger et al., 1983*), and the Obsessive-Compulsive Drinking Scale [OCDS](*Anton et al., 1995*). The Beck Depression Inventory (BDI) is a 21-item self-report inventory designed to measure the severity of depressive symptoms, with a maximum score of 63. The validated French translation of the second version of the BDI (BDI-II) was used in this study (*Bourque and Beaudette, 1982*).

The state report of the State-Trait Anxiety Inventory (STAI Form YA) is a valid and reliable 20-item self-report inventory for measuring the state of anxiety. The scores range from 20 to 80 where higher scores indicate greater anxiety. A valid French version was administered (*Bruchon-Schweitzer and Paulhan, 1993*).

The Obsessive–Compulsive Drinking Scale (OCDS) is a questionnaire that assesses the cognitive aspects of alcohol craving during the preceding 7 days. This 14-question questionnaire provides a global craving score, as well as two subscores: an obsessive score (6 items) and a compulsive score (8 items). A valid French version was used in this study (*Ansseau et al., 2000*).

The amount of alcohol consumed the week before hospitalization was measured in grams per day using the time-line follow back approach (*Sobell and Sobell, 1992*).

## Biological sampling

To avoid variation due to fasting state and circadian rhythm, blood samples were collected in all participants in the morning between 8:00 and 8:30 am after an overnight fasting, at T1 and T2. Blood was drawn in tubes containing EDTA as an anticoagulant. The samples were centrifuged at 1000 × *g* for 15 min at 4 °C and the plasma was frozen at –80 °C until analysis.

## Nontargeted metabolomics analysis

Plasma sample preparation and LC-MS measurement were performed as previously detailed in *Klåvus et al., 2020*. LC-MS analysis. Samples were randomized and thawed on ice before processing. 100 µl of plasma was added to 400 µl of LC-MS grade acetonitrile, mixed by pipetting four time, followed by centrifugation in 700 × *g* for 5 min at 4 °C. A quality control sample was prepared by pooling 10 µl of each sample together. Extraction blanks having only cold acetonitrile and devoid of sample were prepared following the same procedure as sample extracts. LC-MS grade acetonitrile, methanol, water, formic acid, and ammonium formate (Riedel-de Haën, Honeywell, Seelze, Germany) were used to prepare mobile phase eluents in reverse phase (Zorbax Eclipse XDBC18, 2.1×100 mm, 1.8 µm, Agilent Technologies, Palo Alto, CA) and hydrophilic interaction (Acquity UPLC BEH Amide 1.7 µm, 2.1×100 mm, Waters Corporation, Milford, MA) liquid chromatography separation. In reverse phase separation, the samples were analyzed by Vanquish Flex UHPLC system (Thermo Scientific, Bremen, Germany) coupled to high-resolution mass spectrometry (Q Exactive Focus, Thermo Fisher Scientific, Bremen, Germany) in both positive and negative polarity mass range from 120 to 1200, target AGC 1e6 and resolution 70,000 in full scan mode. Data-dependent MS/MS data was acquired for both modes with target AGC 8e3 and resolution 17,500, precursor isolation window was 1.5 amu, normalized collision energies were set at 20, 30, and 40 eV and dynamic exclusion at 10.0 s. In hydrophobic inter-action separation, the samples were analyzed by a 1290 LC system coupled to a 6540 UHD accurate mass Q-ToF spectrometer (Agilent Technologies, Waldbronn, Karlsruhe, Germany) using electrospray ionization (ESI, Jet Stream) in both positive and negative polarity with mass range from 50 to 1600 and scan rate of 1.67 Hz in full scan mode. Source settings were as in the protocol. Data-dependent MS/MS data was acquired separately using 10, 20, and 40 eV collision energy in subsequent runs. Scan rate was set at 3.31 Hz, precursor isolation width of 1.3 amu and target counts/spectrum of 20,000, maximum of 4 precursor pre-cycle, precursor exclusion after 2 spectra and release after 15.0 s. Detectors were calibrated prior sequence and continuous mass axis calibration was performed throughout runs by monitoring reference ions from infusion solution for operating at high accuracy of <2 ppm. Quality control samples were injected in the beginning of the analysis to equilibrate the system and after every 12 samples for quality assurance and drift correction in all modes. All data were acquired in centroid mode by either MassHunter Acquisition B.05.01 (Agilent Technologies) or in profile mode by Xcalibur 4.1 (Thermo Fisher Scientific) softwares.

Metabolomics analysis of TSDS frontal cortex and CSF samples using the same 1290 LC system coupled with a 6540 UHD accurate mass Q-ToF spectrometer has been previously accomplished by *Kärkkäinen et al., 2021*.

Peak picking and data processing. Raw instrumental data (*raw and *.d files) were converted to ABF format using Reifycs Abf Converter (https://www.reifycs.com/AbfConverter). MS-DIAL (Version 4.70) was employed for automated peak picking and alignment with the parameters according to *Klåvus et al., 2020* (*Klåvus et al., 2020*) separately for each analytical mode. For the 6540 Q-ToF mass data minimum peak height was set at 8000 and for the Q Exactive Focus mass data minimum peak height was set at 850,000. Commonly, m/z values up to 1600 and all retention times were considered, for aligning the peaks across samples retention time tolerance was 0.2 min and MS1 tolerance 0.015 Da and the 'gap filling by compulsion' was selected. Alignment results across all modes and sample types as peak areas were exported into Microsoft Excel sheets to be used for further data pre-processing.

Pre-processing including drift correction and quality assessment was done using the notame package v.0.2.1 R software version 4.0.3 separately for each mode. Features present in less than 80% of the samples within all groups and with detection rate in less than 70% of the QC samples were flagged. All features were subjected to drift correction where the features were log-transformed and a regularized cubic spline regression line was fitted for each feature against the quality control samples. After drift correction, QC samples were removed and missing values in the non-flagged features were imputed using random forest imputation. Finally, the preprocessed data from each analytical mode was merged into a single data matrix.

Molecular feature characteristics (exact mass, retention time, and MS/MS spectra) were compared against in-house standard library, publicly available databases such as METLIN, HMDB, and LIPID-MAPS and published literature. Annotation of metabolites and the level of identification was based on the recommendations given by the Chemical Analysis Working Group (CAWG) Metabolomics Standards Initiative (MSI; *Sumner et al., 2007*): 1=identified based on a reference standard, 2=putatively

annotated based on physicochemical properties or similarity with public spectral libraries, 3=putatively annotated to a chemical class and 4=unknown.

## Statistical analysis

R software version 4.0.3. was used for statistical analyses. Multivariate analyses, namely PCA for dimension reduction and sPLS-DA for group discrimination, were conducted by 'mixOmics' R package v. 6.14.1 (*Rohart et al., 2017*). For the sPLS-DA model we used a cross-validation (CV) procedure of 10-fold CV repeated 50 times. Univariate analyses were conducted by 'notame' R package v. 0.2.1 (21). Significant features were shortlisted using Welch's and paired t-tests. All p-values were corrected using the Benjamini-Hochberg false discovery rate (FDR) to calculate the *q*-value. For all tests, p and *q* values <0.05 were considered statistically significant. Visualizations were created by the previously mentioned R packages and GraphPad Prism v. 8.4.2. Correlation analyses were performed at T1 using R software version 3.6.1. Spearman coefficient was calculated and p-value <0.05 was considered statistically significant.

## Results

### Clinical characteristics of the study participants

Two cohorts of sAUD patients (ALCOHOLBIS and GUT2BRAIN) were used in this study. All patients were hospitalized for a 3-week detoxification program, and tested at two timepoints: T1 which represents the first day of alcohol withdrawal and T2 which represents the last day of the detoxification program. Both groups of patients were similar in terms of age, gender, smoking, and drinking habits and presented with high scores of depression, anxiety, and alcohol craving at T1 (*Table 1*). These biological and psychological similarities allow us to combine both cohorts (and consequently increase sample size) and compare them to a group of heathy controls for metabolomics analysis.

### Alterations in the plasma metabolome of sAUD patients

The metabolomics analysis allowed for sorting out a total of 11,651 features from the four analytical modes of the plasma samples. An unsupervised principal component analysis (PCA) model of the plasma metabolomic profiles between healthy controls and sAUD patients at the beginning of the withdrawal (T1) is shown in *Figure 1a*. In addition, the scores plot and the performance of a supervised sparse partial least square discriminant analysis (sPLS-DA) model are shown in (*Figure 1—figure supplement 1* and *Figure 1—figure supplement 2*), respectively. Between healthy controls and sAUD T1, the annotated differential (Welch *t*-test *q*<0.05 and sPLS-DA variable importance in the projection (VIP) score >2.0) metabolites included compounds from several metabolite classes as indicated in *Figure 1b* and *Supplementary file 2*. Compared to healthy controls, the metabolic profiles of sAUD patients were characterized by an increase in long-chain fatty acids, such as 16:1 (palmitoleic acid), 18:1 (octadecenoic acid), and 22:4 (docosatetraenoic acid) fatty acids and phospholipids holding these fatty acids. In addition, several drugs (like diazepam, trazodone) and metabolites with steroid backbone such glycinated bile acids (glycohyodeoxycholic acid and glycochenodeoxycholic acid), steroid hormones and acylcarnitines were increased in the sAUD group. We also observed a significant increase in vitamin B6 metabolite, 4-pyridoxic acid, nicotine metabolite cotinine, a hydroxy fatty acid 3-hydroxyvaleric acid and stress hormone cortisol. However, lysophosphatidylcholines (LPCs) holding a saturated odd-chain (e.g. LPC 15:0 and LPC 17:0), polyunsaturated 18-carbon fatty acid or an ether bond (O-) containing lipid were consistently decreased in sAUD compared to controls (*Figure 1b*). Further, we also showed a decrease in furan fatty acids 3-carboxy-4-methyl-5-pentyl-2-furanpropanoic acid (3-CMPFP) and 3-carboxy-4-methyl-5-propyl-2-furanpropionic acid (CMPF), in a carotenoid compound and in several metabolites belonging to the family of xanthine. In addition, several aminoacid-derived bacterial metabolites such as pipecolic acid, 3-indole propionic acid, p-cresol sulfate, and hippuric acid were significantly decreased in sAUD patients compared to controls. The top-ranked metabolites in *Figure 1b* remained unknown regardless of manual curation.

We then conducted a correlational analysis between blood metabolites and alcohol consumption reported by the patients. Alcohol intake was positively correlated with annotated bile acids, steroids and drugs while xanthines (paraxanthine, theobromine, and theophylline), odd-chain or ether-bond

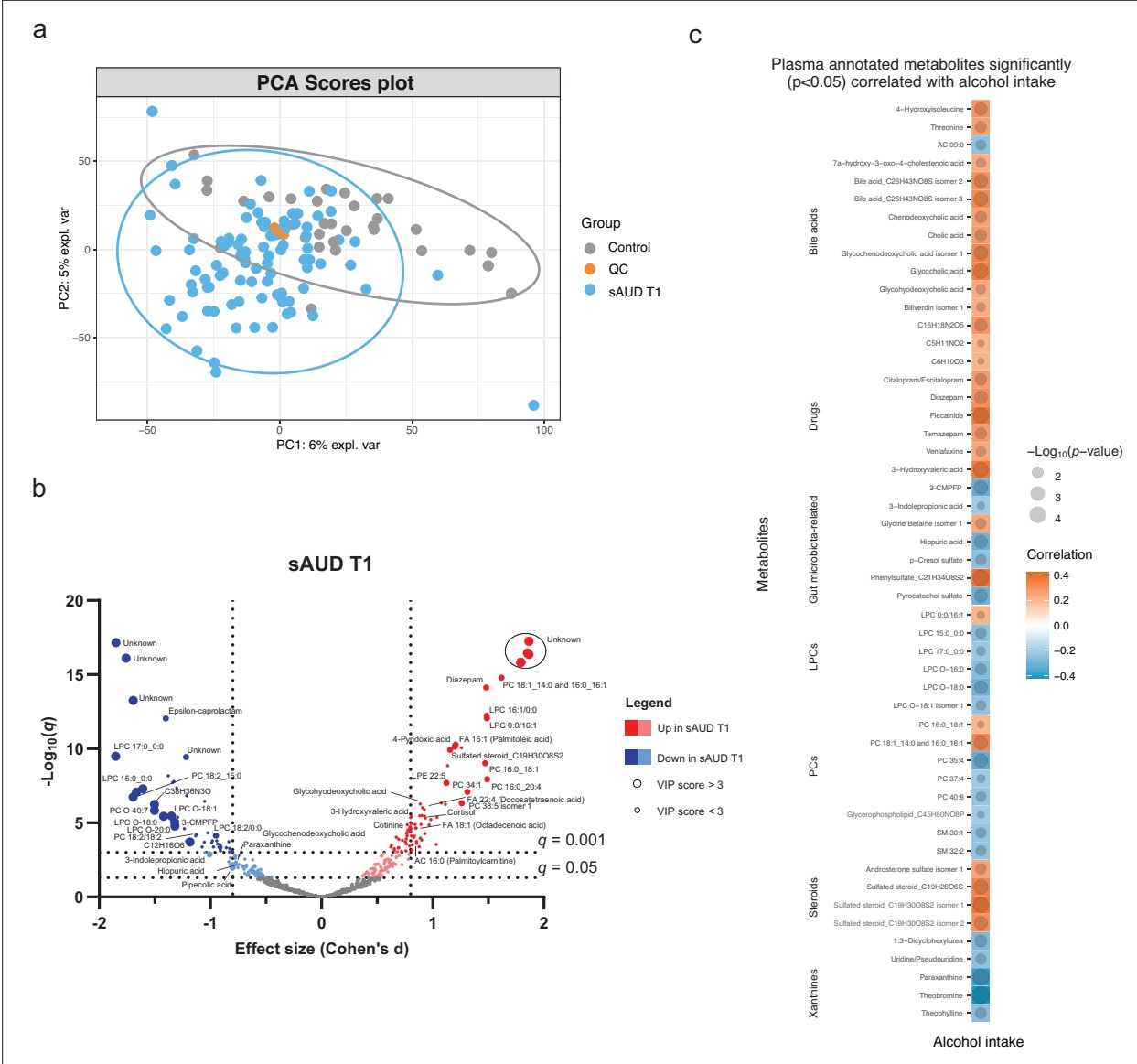

**Figure 1.** Chronic alcohol intake alters the plasma metabolome compared to healthy controls. (**a**) Principal component analysis score plot of the plasma metabolomic features between healthy controls and persons with sAUD at T1. QC samples are colored in orange. (**b**) Volcano plot depicting the effect size (Cohen's D) and -Log$_{10}$ transformed $q$ values derived from Welch's $t$-test analysis of the metabolomic features different between healthy controls and persons with sAUD at T1. Circle size represent the variable importance in projections (VIP) scores derived from the sPLS-DA model for the plasma metabolomic features between persons with sAUD at T1 and healthy controls. (**c**) Annotated metabolites having significant association (Spearman $p<0.05$) with alcohol intake (g/day) in persons with AUD at T1. Circle size refers to the level of significance, blue gradient color to the strength of negative while red to the strength of positive correlation coefficients. 3-CMPFP 3-carboxy-4-methyl-5-propyl-2-furanpropionic acid; AC acylcarnitine; FA fatty acid; LPC lysophosphatidylcholine; LPE lysophosphatidylethanolamine; PC phosphatidylcholine; PE phosphatidylethanolamine; SM sphingomyelin.

The online version of this article includes the following figure supplement(s) for figure 1:

**Figure supplement 1.** Scores plot of the sparse partial least square discriminant analysis (sPLS-DA) separates the plasma metabolome of healthy controls and persons with AUD at the start of the withdrawal (T1).

**Figure supplement 2.** Performance of the sPLS-DA model separating plasma metabolome of healthy controls and persons with AUD at T1.

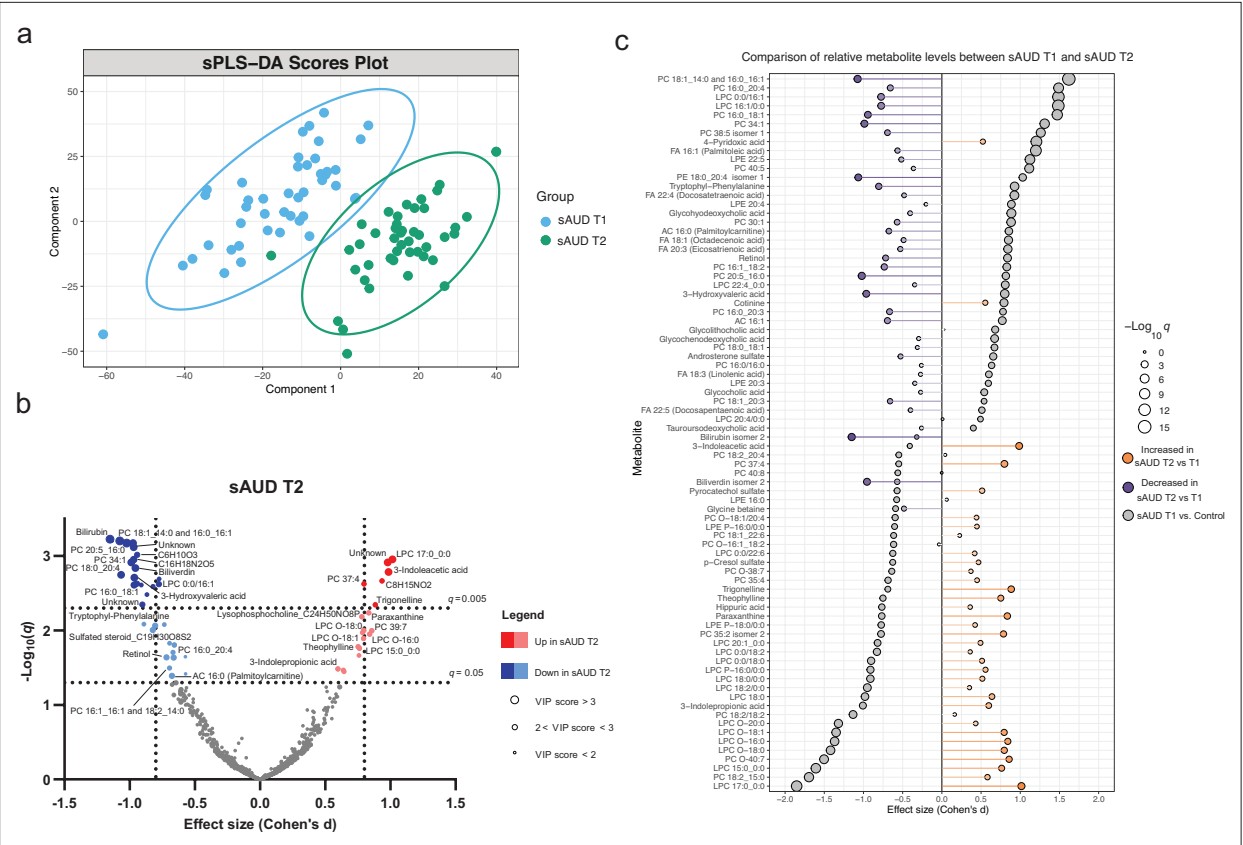

**Figure 2.** 3-week alcohol withdrawal shapes the plasma metabolome. (**a**) Scores plot of the sparse partial least square discriminant analysis (sPLS-DA) of the plasma metabolomes at the start (**T1**) and end (**T2**) of the alcohol withdrawal in persons with sAUD. (**b**) Volcano plot depicting the effect size (Cohen's D) and -Log$_{10}$ transformed $q$ values derived from paired $t$-test analysis of the metabolomic features different between in persons with sAUD at T1 and T2. Circle size represent the variable importance in projections (VIP) scores for the plasma metabolomic features in the sPLS-DA model of persons with sAUD at T1 and T2. (**c**) Lollipop plot of the effect size (Cohen's D) and -Log$_{10}$ transformed $q$ values of the altered annotated metabolites between sAUD T1 and healthy controls as well as sAUD T1 and sAUD T2. Circle size refers to the level of significance, grey color to the comparison between controls and sAUD T1, orange color to relative increase while violet to relative decrease towards the end of alcohol withdrawal (**T2**). AC acylcarnitine; FA fatty acid; LPC lysophosphatidylcholine; LPE lysophosphatidylethanolamine; PC phosphatidylcholine.

The online version of this article includes the following figure supplement(s) for figure 2:

**Figure supplement 1.** Principal component analysis score plot between the plasma metabolomic features in persons with AUD at the beginning (**T1**) and end (**T2**) of alcohol withdrawal.

**Figure supplement 2.** Performance of the sPLS-DA model discriminating plasma metabolome of persons with AUD between the beginning (**T1**) and end (**T2**) of alcohol withdrawal.

**Figure supplement 3.** Changes in dietary intake of coffee, tea, and chocolate during alcohol withdrawal.

LPCs and bacterial metabolite hippuric acid, p-cresol sulfate, pyrocatechol sulfate, and 3-indole propionic acid showed negative correlations (*Figure 1c*).

## Alcohol withdrawal shapes the plasma metabolome

The score plot of the sPLS-DA in *Figure 2a* shows a clear discrimination in plasma metabolomic profiles in the course of withdrawal in sAUD patients. The unsupervised PCA model scores plot and the sPLS-DA model performance are shown in *Figure 2—figure supplement 1* and *Figure 2—figure supplement 2*, respectively. Annotated metabolites discriminating sAUD groups (paired $t$-test $q<0.05$, sPLS-DA VIP score >2.0) before (T1) and after (T2) the 3-week withdrawal period included metabolites from a range of chemical classes (*Figure 2b*, *Supplementary file 2*). Apart from the metabolites belonging to the class of bilirubins, the levels of identified metabolites that were significantly changed upon alcohol withdrawal (the major ones being indoles, acylcarnitines, glycerophospholipids, and xanthines) came back towards to the levels observed in controls (*Supplementary file 2*). In the course

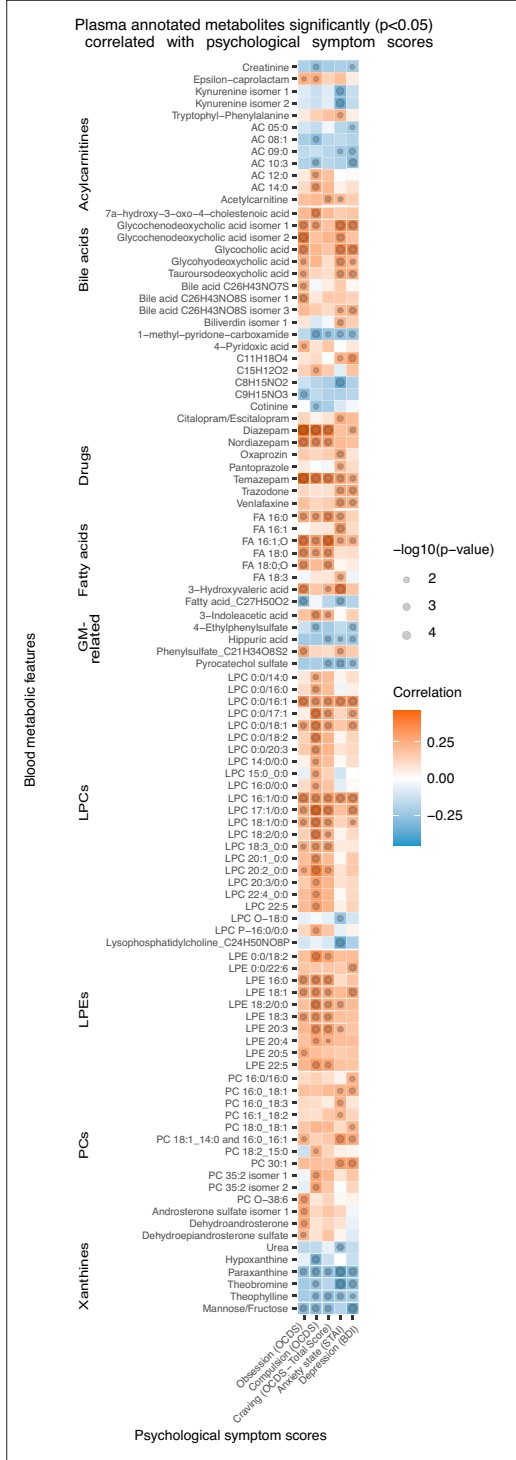

Figure 3. Plasma metabolites associated with psychological symptom scores. Heatmap of the annotated metabolites having significant (p<0.05) Spearman correlation with one or more psychological symptom score of obsession, compulsion, alcohol craving, anxiety state or depression. Circle size refers to the level of significance, blue gradient color to the strength of negative while red to the strength of positive correlation

*Figure 3 continued on next page*

*Figure 3 continued*

coefficients. AC acylcarnitine; FA fatty acid; GM gut microbiota; LPC lysophosphatidylcholine; LPE lysophosphatidylethanolamine; PC phosphatidylcholine.

of alcohol abstinence, we noted a significant decrease in 16-chain acylcarnitines, LPCs with 16- or 18-chain fatty acid tails excluding LPCs with ether bonds, retinol, tryptophyl-phenylalanine dipeptide and 3-hydroxyvaleric acid (*Figure 2b*). On the contrary, LPCs with odd-chain fatty acid tails or ether bonds show a significant increase along with tryptophan derivatives 3-indoleacetic acid and 3-indolepropanoic acid and metabolites of the xanthine family such as theophylline, parax- anthine, theobromine, and trigonelline during alcohol abstinence. The changes in metabolites belonging to the xanthine family during alcohol withdrawal could be explained by the changes in dietary intake of coffee, tea, and chocolate (see *Figure 2—figure supplement 3*).

Overall, *Figure 2c* demonstrates that a number of identified metabolites altered in sAUD patients relative to control are affected by alcohol withdrawal. Apart from 4-pyridoxic acid, cotinine, and heme metabolites bilirubin and biliverdin, the shifts observed in the selected metabolites are generally in the opposite direction as compared to the baseline.

## Correlations between blood metabolites and psychological symptoms

Correlation analysis shows that, at T1, 96 anno- tated features were significantly (p<0.05) correlated with psychological scores of anxiety, depression and alcohol craving (with sub-scores of obsession and compulsion; *Figure 3*). Annotated bile acids, drugs, lysophosphatidylethanolamines (LPE), fatty acids, LPCs apart from LPCs with an ether-bond or 15:0 or 17:0 fatty acid tail were consistently positively correlated with psycho- logical symptoms, and more particularly with the compulsive component of alcohol craving. Members of the xanthine family, pyrocatechol sulfate, a pentose sugar (mannose/fructose), hippuric acid, 1-methyl-pyridone-carboxamide, acylcarnitines with maximum of 10 carbons, creat- inine and kynurenine were negatively correlated with psychological symptoms. Within the acyl- carnitine metabolite class, an interesting pattern was observed, as the shorter chain-length

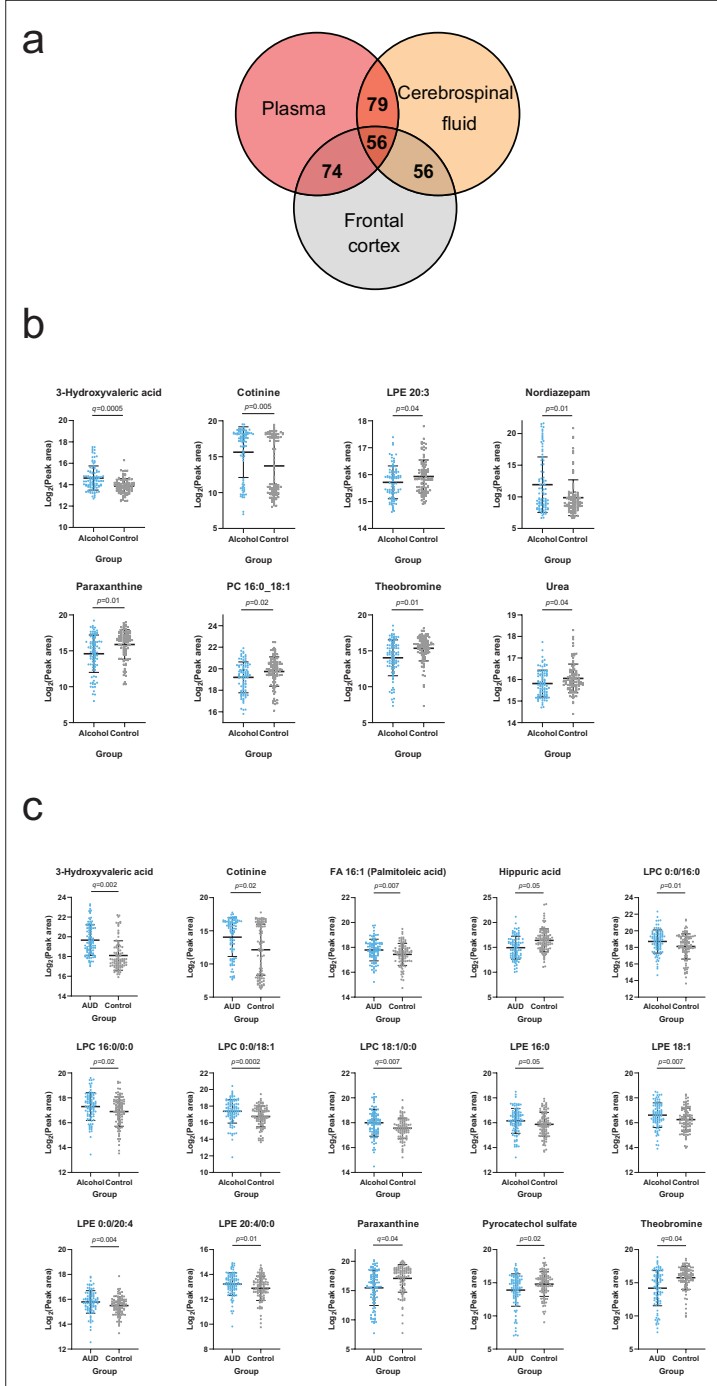

**Figure 4.** Overlapping metabolites within the plasma and brain metabolome. (a) Venn diagram of the annotated plasma metabolites that are also present in the frontal cortex and cerebrospinal fluid metabolome. (b) Significantly altered (p<0.05) frontal cortex metabolites with a corresponding plasma metabolite showing an association with one or more psychological symptom score. (c) Significantly altered (p<0.05) cerebrospinal fluid metabolite with a corresponding plasma metabolite showing an association with one or more psychological symptom score. Data expressed as mean ± SD with individual values shown. Statistical values derived from Welch's t-test comparing metabolomic features between control and alcohol groups.

acylcarnitines were consistently negatively correlated with the psychological parameters, and the long-chain ones demonstrated positive correlation.

## Heavy alcohol-use-related alterations in the brain metabolome

Based on the annotated significantly altered plasma metabolites, we conducted a targeted search in a metabolomics dataset consisting of cerebrospinal fluid (CSF) and frontal cortex samples collected from deceased individuals with a history of heavy alcohol use and control individuals. 79 and 74 of the annotated plasma metabolites were identified from the CSF and frontal cortex, respectively (*Figure 4a*, *Supplementary file 2*). We looked specifically at metabolites significantly correlated with at least one psychological symptom. 3-Hydroxyvaleric acid, cotinine, theobromine and paraxanthine were indeed present in the CNS and significantly (Welch *t*-test p<0.05) different between heavy alcohol use and control groups in both frontal cortex (*Figure 4b*) and CSF (*Figure 4c*). Additional significantly altered metabolites found only in the frontal cortex were LPE 20:3, nordiazepam, PC 16:0_18:1 and urea (*Figure 4b*). In the CSF, the independent differential metabolites were FA 16:1 (palmitoleic acid), hippuric acid, LPCs 16:0 and 18:1, LPEs 16:0, 18:1 and 20:4 and pyrocatechol sulfate (*Figure 4c*).

## Discussion

The circulating metabolome reflects the crosstalk between nutrition, microbiome and host metabolism (*Ahmed et al., 2022*), with diet and microbiome being the strongest determinants of the human blood metabolome (*Bar et al., 2020*; *Chen et al., 2022*). In this study, we showed the impact of sAUD, and the impact of a short-term abstinence, on the blood metabolome. We analyzed the correlations between blood metabolites and psychological symptoms, as well as the presence of identified metabolites in the CNS of individuals considered as heavy alcohol drinkers.

### Impact of sAUD on the blood metabolome

In 2019, a review summarized the results obtained from 23 studies that have used a metabolomics approach for measuring changes in metabolite profiles in relation to alcohol use (*Voutilainen and Kärkkäinen, 2019*). Changes in lipids have been highlighted as the most consistent changes across studies. Lipids are an integral part of cell membranes and signaling molecules in the body. PCs and LPCs have been suggested to form a new class of biomarkers for alcohol consumption (*Jaremek et al., 2013*). For instance, in our study palmitoleic acid (FA 16:1) was largely increased in sAUD patients compared to controls, and in other studies, this metabolite has likewise been significantly associated with alcohol consumption (*Guertin et al., 2014*; *Zheng et al., 2014*). Another clear observation in our study was the lower level of odd-chain lipids in sAUD patients. Since the lipids containing FA 15:0 and FA 17:0 have been suggested to be products from bacterial metabolism (*Hopkins et al., 2001*; *Taormina et al., 2020*), the existence of gut dysbiosis in sAUD patients could explain the lower abundance of LPC 17:0 and LPC 15:0 (*Leclercq et al., 2014*).

We found that some bile acids, sulphated steroids and 3-hydroxyvaleric acid were positively associated with the amount of alcohol consumed. Sulphated steroids and hydroxyvalerate have previously been associated with alcohol intake (*Langenau et al., 2020*; *Wang et al., 2018*). Metabolites belonging to the xanthine family (theobromine, theophylline, paraxanthine) and microbial metabolites (hippuric acid, indole-3-propionic acid, p-cresol sulfate, pyrocatechol sulfate) were negatively correlated with alcohol consumption. Altogether, these results suggest that these metabolites are sensitive to alcohol exposure. Interestingly, these metabolites were also correlated with the severity of the psychological symptoms suggesting that they could play a role in the symptomatology of alcohol use disorder.

### Effect of short-term alcohol abstinence on the blood metabolome

Since alcohol consumption is known to influence lipid metabolism, it was expected that a short-term alcohol abstinence could reverse or ameliorate lipidomic alterations. Indeed, we found that some phospholipids that were increased in sAUD patients at baseline, such as PC 16:0_18:1, PC18:1_14:0 and 16:0_16:1 as well as LPC 16:1, were downregulated during alcohol withdrawal to reach the levels of healthy controls after detox. On the other hand, LPC 15:0 and LPC 17:0 that were decreased

in sAUD patients at baseline, increased during alcohol withdrawal, but did not reach the levels of controls at the end of detox.

The metabolite that contributed the most to the differences observed with alcohol detoxification was bilirubin. While bilirubin was not statistically higher in sAUD patients at baseline vs controls, we observed a significant reduction of this metabolite after a 3-week alcohol withdrawal. In a previous study, serum bilirubin was found to be associated with alcohol consumption, cigarette smoking, and coffee consumption (*Tanaka et al., 2013*). Interestingly, the caffeine metabolites belonging to the xanthine family such as paraxanthine, theophylline, and theobromine that were decreased at baseline in sAUD patients compared to controls, increased significantly during alcohol withdrawal to reach the levels of healthy controls. Changes in dietary intake of coffee, tea, and chocolate during alcohol withdrawal could explain these results. Also, the bacterial metabolites indole derivatives such as 3-indolepropionic acid and 3-indoleacetic acid increased during alcohol withdrawal to reach the levels of healthy controls. Intriguingly, 3-hydroxyvaleric acid significantly decreased during alcohol withdrawal and was found to be lower than healthy controls at the end of detoxification period.

Metabolites that remained significantly higher in sAUD patients at the end of detoxification compared to controls are stress hormone cortisol, palmitoleic acid (FA 16:1), some bile acids, some drugs (diazepam, trazodone), vitB6 metabolite (4-pyridoxic acid, which is likely due to the fact that patients received vitamin B supplements during their hospital stay) and cotinine (nicotine metabolite that reveals the higher proportion of smokers in sAUD patients compared to controls).

## Identification of blood metabolites with potential neuroactive properties

The metabolites belonging to the xanthine family (theobromine, paraxanthine, and theophylline) are metabolites of caffeine produced upon cytochrome-P450-dependent oxidation in the liver. They were all decreased in the blood of sAUD patients at baseline and were negatively correlated with alcohol intake, alcohol craving, depression, and anxiety. The decrease in caffeine metabolites has previously been described in the urine of AUD patients, that is linked to the increasing severity of alcoholic liver disease (*Xu et al., 2023*).

Theobromine is the principle alkaloid found in cocoa beans and is responsible for the bitter taste of chocolate. It is known for its mood improving effect (*Franco and Martínez-Pinilla, 2023*). Like caffeine, theobromine is an inhibitor of brain adenosine receptors and phosphodiesterase. A study in rats showed that the antagonist of A2a adenosine receptor produced a reduction of ethanol reinforcement (*Thorsell et al., 2007*), suggesting adenosine receptor as a potential target for the treatment of alcohol abuse. In a randomized, double-blind, placebo-controlled trial, the phosphodiesterase inhibitor pentoxifylline associated with escitalopram showed greater reduction of depression scores compared to patients receiving escitalopram alone (*El-Haggar et al., 2018*). In another study, Apremilast which is also a phosphodiesterase inhibitor, suppressed excessive alcohol drinking in AUD patients (*Grigsby et al., 2023*). Paraxanthine has a psychostimulant effect and can modulate dopamine release in the striatum (*Orrú et al., 2013*). Interestingly, in 2017 a systematic review indicated that consumption of coffee, tea and cocoa could have protective effects against depression (*García-Blanco et al., 2017*).

Lipids, and mostly LPCs (except ether LPC derivatives) and LPEs were significantly and positively correlated with the compulsive component of alcohol craving. LPCs are secreted by the liver and are actively transported via the blood-brain barrier (BBB) and have been associated with pro-inflammatory events (*Loppi et al., 2018*). LPCs are also precursors of brain lysophosphatidic acid (LPA), which regulates glutamatergic transmission and cortical excitability within the CNS. Recently, LPA has been shown to induce hyperphagia following food restriction and this effect was dependent on hypothalamic agouti-related peptide (AgRP) neurons (*Endle et al., 2022*). AgRP neurons have also been implicated in circuitry controlling non-feeding behavior, including those associated with reward, anxiety and compulsive disorders, more particularly in anorexia nervosa (*Miletta et al., 2020*). Therefore, we hypothesize that the positive correlation between peripheral LPC and compulsion for alcohol drinking found in sAUD patients who have just been deprived of alcohol could be mediated by the effect of LPA on AgRP neurons. Consistent with that, postmortem brain tissues from patients consuming a high intake of alcohol showed increased levels of many LPCs (*Kärkkäinen et al., 2021*).

Circulating bile acids can reach the brain by crossing the BBB, either by simple diffusion or active transport. Some bile acids show neuroprotective effects (*Palmela et al., 2015*) while others are rather neurotoxic (*Quinn et al., 2014*). In Alzheimer disease patients, the levels of glycochenodeoxycholic acid was associated with worse cognition (*MahmoudianDehkordi et al., 2019*). In our study, both primary (glycochenodeoxycholic acid and glycocholic acid) and secondary (glycohyodeoxycholic acid, tauroursodeoxycholic acid) bile acids were positively correlated with depression and anxiety in sAUD patients.

3-Hydroxyvaleric acid, also called β-hydroxypentanoate, was significantly and positively correlated with anxiety and alcohol craving. This metabolite is formed from odd carbon fatty acids in the liver and can reach the brain. 3-Hydroxyvaleric acid is a C5-ketone body and is a precursor of propionyl-CoA that refills intermediates of citric acid cycle and is useful for alternative energy fuel in the brain (*Brunengraber and Roe, 2006*; *Mochel et al., 2005*).

Other co-metabolites, that are produced by the gut microbiota and then processed by the liver, were negatively correlated with all psychological symptoms. Pyrocatechol sulfate is a phenolic compound derived from the gut microbiota, present in the CSF of mice, and implicated in synapse formation and fear extinction learning (*Chu et al., 2019*). In Parkinson disease patients, the plasma level of pyrocatechol sulfate is decreased compared to controls (*Chen and Lin, 2022*). In our study, blood pyrocatechol sulfate was significantly and negatively correlated with all psychological symptoms of sAUD patients (i.e. anxiety, depression, and craving) suggesting a neuroprotective role of this metabolite. Interestingly, 4-ethylphenylsulfate, another gut-derived metabolite linked with neurodevelopment abnormalities, autism and anxiety behavior in mice (*Needham et al., 2022*; *Hsiao et al., 2013*; *Needham et al., 2021*) showed negative correlations with depression and the compulsive component of alcohol craving.

Hippuric acid, the glycine conjugate of benzoic acid has long been associated with the microbial degradation of specific dietary components, including polyphenolic compounds (like chlorogenic acid and catechin) found in fruits, vegetables, coffee, and tea (*Lees et al., 2013*). Hippuric acid is indeed a host-microbe cometabolite (*Pruss et al., 2023*). It is synthesized in the liver and in the renal cortex from the microbial metabolite benzoate. The plasma concentration of hippuric acid has been shown to be 17-fold higher in conventional mice compared with their germ-free counterparts suggesting a substantial contribution of the gut microbiota in its production (*Wikoff et al., 2009*). We showed that blood hippurate levels, that correlated negatively with anxiety, depression and craving, were decreased in sAUD patients, as shown in ethanol-treated mice (*Gao et al., 2011*) and humans characterized by high alcohol intake and those with major depression (*Harada et al., 2016*; *Kärkkäinen et al., 2024*). Urinary hippurate excretion is also decreased in depression, schizophrenia, and autism spectrum disorders patients (*Lees et al., 2013*). A recent Mendelian randomization study including >13,000 individuals from five European cohorts characterized for depression suggested that low hippuric acid levels in the circulation is part of the causal pathway leading to depression (*van der Spek et al., 2023*), which was consistent with a significant decrease of the dietary sources of hippuric acid including fresh fruits and vegetables in depressed patients (*van der Spek et al., 2023*).

Another way to support the neuroactive effects of the blood metabolites that are correlated with one or several psychological factors is to demonstrate their presence in the brain. We therefore conducted a targeted search in a database of post-mortem frontal cortex and CSF metabolomics analysis (*Kärkkäinen et al., 2021*) and found that 3-hydrovaleric acid, caffeine metabolites (theobromine, paraxanthine, and theophylline) and microbial metabolites (hippuric acid and pyrocatechol sulfate) that were correlated with anxiety, depression and alcohol craving in our plasma cohort were also present in the brain and in CSF, and the direction of their changes in the plasma (increase or decrease) mimicked changes in the central nervous system.

## Advantages and limitations of the study

Most of the studies assessing the impact of alcohol consumption on the blood metabolome were cross-sectional, and only included male participants (*Harada et al., 2016*). Here, we reported longitudinal data to assess the impact of a short-term alcohol abstinence on the blood metabolome, both in male and female AUD patients. In the study of *Zhu et al., 2021*, the AUD patients recruited were alcohol abstinent, but for various periods of abstinence. To avoid the bias of abstinence duration, our patients were enrolled in a rigorous and standardized manner, within 24 hr after the last drink.

Furthermore, alcohol consumption was carefully evaluated with the time line follow back method, which allows precise calculation of the amount (and type) of alcohol consumed (*Sobell and Sobell, 1992*).

This study also presents some limitations. First, the metabolomics analysis was conducted with LC-MS while some important molecules, like lipoproteins, could have been measured with NMR-based methods. Combining NMR and MS-based methods could have covered a wider spectrum of metabolites. However, the non-targeted metabolic profiling with two different chromatographic methods and ionization polarities covers a wide range of metabolites ideal for our discovery-based approach. Large studies are usually required in metabolomics to observe small and medium size changes. Here, we included only s96 AUD patients, but they were all well characterized and received standardized therapies (for instance, vitB supplementation) during alcohol withdrawal.

The selection of the control group is always challenging in alcohol research. Here, the healthy subjects were matched for sex, age, and BMI but not for smoking status or nutritional intake. Alcohol addiction is a major cause of malnutrition in developed countries and tobacco smoking is more prevalent in alcohol users compared to healthy subjects. These two main confounding factors, although being an integral part of the alcoholic pathology, are known to influence the blood metabolome (*Hsu et al., 2017*; *Barve et al., 2017*; *Harrison et al., 2008*). Furthermore, another limitation is that the control group was tested only once, while the sAUD patients were tested twice (T1 and T2). This means that we do not take into consideration the intra-personal variability of the metabolomics data when interpreting the results of alcohol withdrawal effects.

## Conclusion

LC-MS metabolomics plasma analysis allowed for the identification of metabolites that were clearly linked to alcohol consumption, and reflected changes in metabolism, alterations of nutritional status, and gut microbial dysbiosis associated with alcohol intake. In particular the changes in lipid class involving odd-chain fatty acids and ether-bond lipids as well as compounds produced by gut microbiota seem to be the most prominent indicators of metabolic malfunction related to severe alcohol use disorder, and thus warrant further studies and targeted intervention. Also, the discovery of metabolites associated with behavioral and psychiatric traits related to sAUD were of importance, and could be considered potential new therapeutic targets in the management of sAUD, namely as adjuvants in the period of alcohol abstinence. The novelty of our work was to characterize the impact of sAUD on the blood metabolome, and the impact of a short-term alcohol abstinence in the same individuals, within a cohort that included both male and female patients. Intervention studies are needed in order to bring the proof of concept that nutritional approaches – namely the addition of specific lipids, or of nutrients modulating the gut microbiome - for example prebiotic dietary fibers - may be essential and so far underestimated components of alcohol withdrawal efficacy.

## Acknowledgements

SL is a Research Associate of the Fonds de la Recherche Scientifique – FNRS. Metabolomics analysis of the TSDS samples was supported by grant from the Finnish Foundation for Alcohol Studies. PDT received funding from Fondation Saint Luc. NMD is a recipient of grants from the Fonds de la Recherche Scientifique (FRS-FNRS) [PDR T.0068.19], and from the Fédération Wallonie-Bruxelles (Action de Recherche Concertée ARC18- 23/092).

## Additional information

### Competing interests
Olli K Kärkkäinen, Kati Hanhineva: Founder of Afekta Technologies Ltd. The other authors declare that no competing interests exist.

## Funding

| Funder | Grant reference number | Author |
| --- | --- | --- |
| Research Council of Finland | | Kati Hanhineva |
| Agence Nationale de la Recherche | ANR-19-NEUR-0003-03 | Sophie Laye |
| Fonds De La Recherche Scientifique - FNRS | R.8013.19 | Nathalie Delzenne |

The funders had no role in study design, data collection and interpretation, or the decision to submit the work for publication.

## Author contributions

Sophie Leclercq, Conceptualization, Formal analysis, Investigation, Visualization, Writing – original draft; Hany Ahmed, Data curation, Formal analysis, Visualization, Methodology; Camille Amadieu, Data curation, Formal analysis, Investigation; Géraldine Petit, Marie Poncin, Investigation; Ville Koistinen, Quentin Leyrolle, Formal analysis; Peter Stärkel, Conceptualization, Resources, Project administration; Eloise Kok, Collected and provided the TSDS post-mortem samples; Pekka J Karhunen, Collected and provided the TSDS post-mortem samples; Philippe de Timary, Conceptualization, Project administration; Sophie Laye, Data curation; Audrey M Neyrinck, Data curation, Formal analysis, Project administration; Olli K Kärkkäinen, Data curation, Formal analysis; Kati Hanhineva, Resources, Data curation, Supervision, Funding acquisition, Writing – review and editing; Nathalie Delzenne, Conceptualization, Supervision, Validation, Project administration, Writing – review and editing

## Author ORCIDs

Sophie Leclercq ⓘ http://orcid.org/0000-0002-2894-5220
Quentin Leyrolle ⓘ https://orcid.org/0000-0002-0763-1433
Eloise Kok ⓘ https://orcid.org/0000-0001-7399-9749
Audrey M Neyrinck ⓘ http://orcid.org/0000-0002-9435-3338
Olli K Kärkkäinen ⓘ https://orcid.org/0000-0003-0825-4956
Nathalie Delzenne ⓘ https://orcid.org/0000-0003-2115-6082

## Ethics

Clinical trial registration NCT03803709 at ClinicalTrials.gov.

The study was approved by the "Comité d'éthique Hospitalo-facultaire Saint-Luc UCLouvain" (2017/04JUL/354 and 2014/31dec/614, identification number NCT03803709 at ClinicalTrials.gov). The study has been carried out in accordance with The Code of Ethics of the World Medical Association and followed the ethical guidelines set out in the Declaration of Helsinki. All participants provided written informed consent in compliance with the European law 2001/20/CE guidelines.

Reviewer #1 (Public review): https://doi.org/10.7554/eLife.96937.3.sa1
Reviewer #2 (Public review): https://doi.org/10.7554/eLife.96937.3.sa2
Reviewer #3 (Public review): https://doi.org/10.7554/eLife.96937.3.sa3
Author response https://doi.org/10.7554/eLife.96937.3.sa4

# Additional files

## Supplementary files

• Supplementary file 1. biological features of healthy controls and background characteristics of the selected subjects from the TSDS cohort.

• Supplementary file 2. List of identified metabolites and unknown features and related statistics.

• Supplementary file 3. EICs, retention times and reference spectra for level 1 identifications.

• MDAR checklist

## Data availability

Most of the data generated or analysed during this study are included in the manuscript and supporting files: background characteristics and clinical features of the selected subjects from the different cohorts are presented in Table 1 and Supplementary File 1; analytical characteristics and statistical outputs for all identified metabolites are presented Supplementary File 2; Extracted ion chromatograms (EIC) of identified metabolites are presented in Supplementary File 3. Psychological and metabolic parameters analysed in the GUT2BRAIN cohort are presented in our previous works (*Amadieu et al., 2022a*; *Amadieu et al., 2022b*). Any additional information required to reanalyze the data reported in this paper is available from the lead contacts upon request.

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
