## [Editor Report · eLife assessment]

This study provides **valuable** insights and allows for hypothesis generation around diet-microbe-host interactions in alcohol use disorder. The strength of the evidence is **convincing**: the work is done in a rigorous manner in a well-described cohort of patients with AUD before and after withdrawal. There are several weaknesses, including validating the metabolites identified by metabolomics, the cross-sectional study design, the lack of a healthy control group, and the descriptive nature of such clinical cohort studies. Nevertheless, the study provides a wealth of new data that may be the basis for future studies that test causality and elucidate the role of single metabolites in the psychiatric sequela of AUD.

---

## [Referee Report · Reviewer #1 (Public review)]

Summary:

This work by Leclercq and colleagues performed metabolomics on biospecimens collected from 96 patients diagnosed with severe alcohol use disorder (AUD). The authors discovery strong alterations in circulating glycerophospholipids, bile acids, and some gut microbe-derived metabolites in AUD patients compared to controls. An exciting part of this work is that metabolomics was also done in post-mortem samples of the frontal cortex and cerebrospinal fluid of heavy alcohol users, and some of the same metabolites were seen to be altered in the central nervous system. This important study will form the basis for hypothesis generation around diet-microbe-host interactions in alcohol use disorder. The work is done in a highly rigorous manner, and the rigorously collected human samples is an evident strength of this work. Overall, this work will provide many new insights, and it is poised to have a high impact on the field.

Strengths:

(1) The rigorously collected patient-derived samples

(2) There is high rigorous in the metabolomics investigation

(3) Statistical analyses are well-described and strong.

(4) The careful control of taking blood samples at the same time to avoid alterations in meal- and circadian-related fluctuations in metabolites is a clear strength.

Weaknesses:

None remaining

---

## [Referee Report · Reviewer #2 (Public review)]

The authors carried out the current studies with the justification that the biochemical mechanisms that lead to alcohol addiction are incompletely understood. The topic and question addressed here are impactful and indeed deserve further research. To this end, a metabolomics approach toward investigating the metabolic effects of alcohol use disorder and the effect of alcohol withdrawal in AUD subjects is valuable. However, this work is primarily descriptive in nature, and these data alone do not meet the stated goal of investigating biochemical mechanisms of alcohol addiction. The current work's most significant limitation is the cross-sectional study design, though inadequate description and citation of the underlying methodological approaches also hampers interest.

Most of the data are cross-sectional in study design, i.e., alcohol use disorder vs controls. However, it is well established that there is a high degree of interpersonal variation with metabolism, and further, there is somewhat high intra-personal variation in metabolism over time. This means that the relatively small cohort of subjects is unlikely to just reflect the broader condition of interest (AUD/withdrawal). The authors report a comparison of a later time-point after alcohol withdrawal (T2) vs the AUD condition. Nonetheless, without replicate time points from the control subjects it is difficult to assess how much of these changes are due to withdrawal vs the intra-personal variation described above. Overall, insufficient experimental context exists to interpret these findings into a biological understanding. For example, while several metabolites are linked with AUD and associated with microbiome or host metabolism based on existing literature, it is unclear from the current study what function these changes have concerning AUD, if any. The authors also argue that alcohol withdrawal shifts the AUD plasma metabolic fingerprint towards healthy controls (line 153). However, this is hard to assess based on the provided plots since the direction of change of the orange data subset considers AUD T2 vs. T1. In contrast, AUD T2 vs. Control would represent the claimed shift. To substantiate these claims, the authors would better support their argument by showing this comparison in all experimental groups (including control subjects) in their multi-dimensional model (e.g., PCA). The authors attempt to extend the significance of their findings by assessing post-mortem brain tissues from AUD subjects; however, the finding that many of the metabolites changed in T2/T1 are also found in AUD brain tissues is interesting but does not strongly support the authors' claims that these metabolites are markers of AUD (line 173). Concerning the plasma cohort itself, it is unclear how the authors assessed for compliance with alcohol withdrawal or whether the subjects' blood-alcohol levels were independently verified.

The second area of concern is the lack of description of the analytical methodology, the lack of metabolite identification validation evidence, and related statistical questions. The authors cite reference #59 regarding the general methodology. However, this reference from their group is a tutorial/review/protocol focused resource paper and it needs to be clarified how specific critical steps were actually applied to the current plasma study samples, given the range of descriptions provided in the citations. The authors report a variety of interesting metabolites, including their primary fragment intensities, which is appreciated (Supp Table 3), but no MS2 matching scores are provided for level 2 or 3 hits. Further, level 1 hits under their definition are validated by an in-house standard, but not supporting data are provided other than this categorization. Finally, a common risk in such descriptive studies is finding spurious associations, especially considering the many factors as described in the current work. These include AUD, depression, anxiety, craving, withdrawal, etc. The authors describe the use of BH correction for multiple-hypothesis testing. Still, this approach only accounts for the many possible metabolite association tests within each comparison (such as metabolites vs. depression) and does not account for the multi-variate comparisons to the many behavior/clinical factors described above. The authors should employ one of several common strategies, such as linear mixed effects models for these types of multi-variate assessments.

Revised Review after Resubmission:

I thank the authors for their responses and revisions to the figures and data and their clarifications of their results and study goals. However, based on this updated information, it is now more apparent that the paper falls into the common trap of descriptive studies where insufficient experimental design was considered to test the association in question robustly. Further, follow-up initiatives are lacking to test the findings by other experimental means. Despite the authors' responses, the paper still fails to convert or interpret the metabolomics findings into any new biological understanding or meaningfully testable hypotheses, and the results remain descriptive in nature with significant caveats.

The authors clarify that their study's "goal was not to investigate the biochemical mechanisms of AUD but how metabolomics could contribute to the psychological alterations of AUD." However, the 2nd sentence of the abstract remains as follows: "The biochemical mechanisms that lead to this disorder are not yet fully understood, and in this respect, metabolomics represents a promising approach to decipher metabolic events related to AUD."This leads the reader to conclude that the purpose of the current study is to use metabolomics to address this gap, despite their later clarification. In the revised response, the authors walk back their claims of these goals, yet the manuscript text and data is largely unchanged in the revision. The serious caveats pointed out by several reviewers concerning the study as reported significantly reduces the utility of the described findings for the broader scientific community, and the authors largely downplay these limitations without addressing the underlying issues.

The authors also clarified in their response that the study's key purpose of the study is to assess "correlations between the blood metabolome and psychological symptoms developed in AUD patients." This goal is dubious as the vast majority of metabolites are not psychoactive, and it is implausible that the metabolome would affect mental state or vice versa. More biological frameworks and citations are needed for this paradigm. The soundness of the goal is further questioned by the study's simplistic design and the authors' admission that "In this discovery-based approach, the aim was to discover potential candidates linked with psychological symptoms for subsequent work to evaluate causality." Yet, the authors side-step the point about the risk of finding spurious associations and decline to control this risk using widely-accepted approaches such as multi-variate correction, instead continuing to use only BH correction for multiple hypothesis testing. The reviewers previously pointed out that BH correction only accounts for the many possible metabolite association tests within each comparison (such as metabolites vs depression). However, it does not account for the multi-variate comparisons to the many behavior/clinical factors. This issue is ignored in the response because the study's goal is hypothesis generating. Instead, the authors focused their responses on the issue of causality which was not the central point of the criticism.

Further, the authors employ mainly systemic plasma analyses unlikely to reflect brain biochemistry. The authors deny that the purpose of including the post-mortem brain tissue data was to demonstrate that "metabolites significantly correlated with the psychological symptoms - and present in the central nervous system (frontal cortex or CSF) - are "markers of AUD," yet if this is not the goal, the structure of the experiment, and the value of these data, is unclear. Another reviewer pointed out that it is difficult to control cross-sectional post-mortem tissue due to a lack of suitable controls, and the authors again side-step the question by citing the lack of suitable controls and the impossibility of "healthy controls" in post-mortem samples. This is true, but this lack of technical feasibility and the confounding factor of CVD/lipid metabolism does not justify the weak experimental design in this respect. Therefore, it remains unclear what can be understood from these data, given the limitations.

Finally, the authors acknowledge the limitation in their revision that they did not assess a second-time point in the control cohort of samples which could have been used to tease apart intra-personal variation from AUD-associated changes during alcohol-abstinence. Unfortunately, this is not a small caveat to simply acknowledge in the discussion section; it severely limits the interpretation and utility of the reported data more broadly, and the authors do not address this underlying problem.

---

## [Referee Report · Reviewer #3 (Public review)]

Summary:

The authors have compared different groups of AUD patients at different levels and have examined metabolomics.

Strengths:

A well-written and comprehensive study.

---

## [Author Response]

The following is the authors’ response to the original reviews.

**Public reviews:**

**Reviewer #1:**
This work by Leclercq and colleagues performed metabolomics on biospecimens collected from 96 patients diagnosed with several types of alcohol use disorders (AUD). The authors discovered strong alterations in circulating glycerophospholipids, bile acids, and some gut microbe-derived metabolites in AUD patients compared to controls. An exciting part of this work is that metabolomics was also performed in frontal cortex of post-mortem brains and cerebrospinal fluid of heavy alcohol users, and some of the same metabolites were seen to be altered in the central nervous system. This is an important study that will form the basis for hypothesis generation around diet-microbe-host interactions in alcohol use disorder. The work is done in a highly rigorous manner, and the rigorously collected human samples are a clear strength of this work. Overall, many new insights may be gained by this work, and it is poised to have a high impact on the field.Strengths:(1) The rigorously collected patient-derived samples.(2) There is high rigor in the metabolomics investigation.(3) Statistical analyses are well-described and strong.(4) An evident strength is the careful control of taking blood samples at the same time of the day to avoid alterations in meal- and circadian-related fluctuations in metabolites.Weaknesses:(1) Some validation in animal models of ethanol exposure compared to pair-fed controls would help strengthen causal relationships between metabolites and alterations in the CNS.(2) The classification of "heavy alcohol users" based on autopsy reports may not be that accurate.(3) The fact that most people with alcohol use disorder choose to drink over eating food, there needs to be some more discussion around how dietary intake (secondary to heavy drinking) most likely has a significant impact on the metabolome.

We thank this reviewer for his/her encouraging comments and for highlighting the fact that this study is important in the field to generate hypotheses around diet-microbe-host interactions in alcohol use disorder.

Concerning weakness #1: Regarding the validation in animal models of ethanol exposure, we were very careful in our discussion to avoid pretending that the study allowed to test causality of the factors. This was certainly not the objective of the present study. The testing of causality would indeed probably necessitate animal models but these models could only test the effects of one single metabolite at a time and could not at the same time capture the complexity of the changes occurring in AUD patients. The testing of metabolites would be a totally different topic. Hence, we do not feel comfortable in conducting rodent experiments for several reasons. First, AUD is a very complex pathology with physiological and psychological/psychiatric alterations that are obviously difficult to reproduce in animal models. Secondly, as mentioned by the reviewer, AUD pathology spontaneously leads to nutritional deficits, including significant reductions in carbohydrates, lipids, proteins and fiber intakes. We have recently published a paper in which we carefully conducted detailed dietary anamneses and described the changes in food habits in AUD patients (Amadieu et al., 2021). As explained below, some blood metabolites that are significantly correlated with depression, anxiety and craving belong to the xanthine family and are namely theobromine, theophylline, and paraxanthine, which derived from metabolism of coffee, tea or chocolate (which are not part of the normal diet of mice or rats).Therefore, conducting an experiment in animal model of ethanol exposure compared to pair-fed controls will omit the important impact of nutrition in blood metabolomics and consequently won’t mimic the human AUD pathology. In addition, if we take into consideration the European Directive 2010/63/EU (on the protection of animals used for scientific purposes) which aims at Reducing (Refining, Replacing) the number of animals used in experiment, it is extremely difficult to justify, at the ethical point of view, the need to reproduce human results in an animal model that won’t be able to mimic the nutritional, physiological and psychological alterations of alcohol use disorder.

Concerning weakness #2: The classification of subjects to the group who have a history of heavy alcohol use was not solely on autopsy record, but was also based on medical history i.e. diagnosis of alcohol-related diseases: ICD-10 codes F10.X, G31.2, G62.1, G72.1, I42.6, K70.0-K70.4, K70.9, and K86.0, or signs of heavy alcohol use in the clinical or laboratory findings, e.g., increased levels of gamma-glutamyl transferase, mean corpuscular volume, carbohydrate-deficient transferrin, as stated in the methods section of the manuscript. In Finland, the medical records from the whole life of the subjects are available. We consider that getting diagnosis of alcohol-related disease is clear sign of history of heavy alcohol use.

Concerning weakness#3: As explained above, we do agree with the reviewer that AUD is not only “drinking alcohol” but is also associated with reduction in food intake that obviously influenced the metabolomics data presented in this current study. We have therefore added some data, which have not been published before, in the results section that refer to key nutrients modified by alcohol intake and we refer to those data and their link with metabolomics in the discussion section:

Results section page 8, Line 153-155. This sentence has been added:

“The changes in metabolites belonging to the xanthine family during alcohol withdrawal could be explained by the changes in dietary intake of coffee, tea and chocolate (see Fig S5).”

Discussion section: Page 11, Line 235-240.

“Interestingly, the caffeine metabolites belonging to the xanthine family such as paraxanthine, theophylline and theobromine that were decreased at baseline in AUD patients compared to controls, increased significantly during alcohol withdrawal to reach the levels of healthy controls. Changes in dietary intake of coffee, tea and chocolate during alcohol withdrawal could explain these results”.

In the conclusion, Page 16, Line 354-356, we clearly stated that: “LC-MS metabolomics plasma analysis allowed for the identification of metabolites that were clearly linked to alcohol consumption, and reflected changes in metabolism, alterations of nutritional status, and gut microbial dysbiosis associated with alcohol intake”

Reference:

Amadieu C, Leclercq S, Coste V, Thijssen V, Neyrinck AM, Bindels LB, Cani PD, Piessevaux H, Stärkel P, Timary P de, Delzenne NM. 2021. Dietary fiber deficiency as a component of malnutrition associated with psychological alterations in alcohol use disorder. *Clinical Nutrition*
**40**:2673–2682. doi:10.1016/j.clnu.2021.03.029

Leclercq S, Cani PD, Neyrinck AM, Stärkel P, Jamar F, Mikolajczak M, Delzenne NM, de Timary P. 2012. Role of intestinal permeability and inflammation in the biological and behavioral control of alcohol-dependent subjects. *Brain Behav Immun*
**26**:911–918. doi:10.1016/j.bbi.2012.04.001

Leclercq S, De Saeger C, Delzenne N, de Timary P, Stärkel P. 2014a. Role of inflammatory pathways, blood mononuclear cells, and gut-derived bacterial products in alcohol dependence. *Biol Psychiatry*
**76**:725–733. doi:10.1016/j.biopsych.2014.02.003

Leclercq S, Matamoros S, Cani PD, Neyrinck AM, Jamar F, Stärkel P, Windey K, Tremaroli V, Bäckhed F, Verbeke K, de Timary P, Delzenne NM. 2014b. Intestinal permeability, gut-bacterial dysbiosis, and behavioral markers of alcohol-dependence severity. *Proc Natl Acad Sci U S A*
**111**:E4485–E4493. doi:10.1073/pnas.1415174111

Voutilainen T, Kärkkäinen O. 2019. Changes in the Human Metabolome Associated With Alcohol Use: A Review. *Alcohol and Alcoholism*
**54**:225–234. doi:10.1093/alcalc/agz030

**Public Reviewer #2:**
The authors carried out the current studies with the justification that the biochemical mechanisms that lead to alcohol addiction are incompletely understood. The topic and question addressed here are impactful and indeed deserve further research. To this end, a metabolomics approach toward investigating the metabolic effects of alcohol use disorder and the effect of alcohol withdrawal in AUD subjects is valuable. However, it is primarily descriptive in nature, and these data alone do not meet the stated goal of investigating biochemical mechanisms of alcohol addiction. The current work's most significant limitation is the cross-sectional study design, though inadequate description and citation of the underlying methodological approaches also hampers interest. Most of the data are cross-sectional in the study design, i.e., alcohol use disorder vs controls. However, it is well established that there is a high degree of interpersonal variation with metabolism, and further, there is somewhat high intra-personal variation in metabolism over time. This means that the relatively small cohort of subjects is unlikely to reflect the broader condition of interest (AUD/withdrawal). The authors report a comparison of a later time-point after alcohol withdrawal (T2) vs. the AUD condition. However, without replicative time points from the control subjects it is difficult to assess how much of these changes are due to withdrawal vs the intra-personal variation described above.

We agree with the reviewer. Our goal was not to investigate the biochemical mechanisms of AUD but rather to investigate how metabolomics could contribute to the psychological alterations of AUD. The goals of the study are defined at the end of the introduction (Page 4 – Lines 80-91), as follows:

“The aims of this study are multiple. First, we investigated the impact of severe AUD on the blood metabolome by non-targeted LC-MS metabolomics analysis. Second, we investigated the impact of a short-term alcohol abstinence on the blood metabolome followed by assessing the correlations between the blood metabolome and psychological symptoms developed in AUD patients. Last, we hypothesized that metabolites significantly correlated with depression, anxiety or alcohol craving could potentially have neuroactive properties, and therefore the presence of those neuroactive metabolites was confirmed in the central nervous system using post-mortem analysis of frontal cortex and cerebrospinal fluid of persons with a history of heavy alcohol use. Our data bring new insights on xenobiotics- or microbial-derived neuroactive metabolites, which can represent an interesting strategy to prevent or treat psychiatric disorders such as AUD”.

Due to the fact that the method section describing the study design is located at the end of the manuscript, we have decided to clarify the methodological approach in the first paragraph of the result section in order to show that in fact, we have performed a longitudinal study (which includes the same group of AUD, tested at two time points – at the beginning and at the end of alcohol withdrawal). This is stated as follows:

Results section, Page 6, Line 97-99: “All patients were hospitalized for a 3-week detoxification program, and tested at two timepoints: T1 which represents the first day of alcohol withdrawal, and T2 which represents the last day of the detoxification program”.

We propose to add a figure with a schematic representation of the protocol. We let the editor deciding whether this figure can be added (as supplemental material).

**Author response image 1. sa4fig1:** Schematic representation of the protocol.

We agree with the reviewer that the correlational analysis (between blood metabolites and psychological symptoms) is conducted at one time point (T1) only, which has probably led to the confusion between cross-sectional and longitudinal study. In fact we had a strong motivation to provide correlations at T1, instead of T2. T1, which is at the admission time, is really the moment where we can take into account variability of the psychological scores. Indeed, after 3 weeks of abstinence (T2), the levels of depression, anxiety and alcohol craving decreased significantly (as shown in other studies from our group Leclercq et al., 2014b, 2014a, 2012) and remained pretty low in AUD patients, with a much lower inter-individual variability which makes the correlations less consistent.

We agree with the reviewer that there is a high intra and inter-personal variability in the metabolomics data, that could be due to the differences in previous meals intakes within and between subjects. While AUD subjects have been tested twice (at the beginning and at the end of a 3-week detoxification program), the control subjects have only been tested once. Consequently, we did not take into account the intra-personal variability in the control group. The metabolomics changes observed in AUD patients between T1 and T2 are therefore due to alcohol withdrawal but also to intra-personal variability. This is a limitation of the study that we have now added in the discussion section, Page 16, Lines 354-357 as follows:

“The selection of the control group is always challenging in alcohol research. Here, the healthy subjects were matched for sex, age and BMI but not for smoking status or nutritional intake. Alcohol addiction is a major cause of malnutrition in developed countries and tobacco smoking is more prevalent in alcohol users compared to healthy subjects. These two main confounding factors, although being an integral part of the alcoholic pathology, are known to influence the blood metabolome. Furthermore, another limitation is that the control group was tested only once, while the AUD patients were tested twice (T1 and T2). This means that we do not take into consideration the intra-personal variability of the metabolomics data when interpreting the results of alcohol withdrawal effects”.

The limitation concerning the small sample size is already mentioned in the discussion section, as follows:

“Large studies are usually required in metabolomics to observe small and medium size changes. Here, we included only 96 AUD patients, but they were all well characterized and received standardized therapies (for instance, vitB supplementation) during alcohol withdrawal”.

Overall, there is not enough experimental context to interpret these findings into a biological understanding. For example, while several metabolites are linked with AUD and associated with microbiome or host metabolism based on existing literature, it's unclear from the current study what function these changes have concerning AUD, if any. The authors also argue that alcohol withdrawal shifts the AUD plasma metabolic fingerprint towards healthy controls (line 153). However, this is hard to assess based on the plots provided since the change in the direction of the orange data subset is considers AUD T2 vs T1. In contrast, AUD T2 vs Control would represent the claimed shift. To support these claims, the authors would better support their argument by showing this comparison as well as showing all experimental groups (including control subjects) in their multi-dimensional model (e.g., PCA).

We thank the reviewer for these comments. It is true in this type of discovery-based approach the causality cannot be interpreted nor do we claim so. The aim was to characterize the metabolic alterations in this population, response to withdrawal period and suggest potential candidate metabolites linked to psychological symptoms. Rigorous pre-clinical assays and validation trials in humans are required to prove the causality, if any, of the discussed metabolites.

The original claim on line 153 was poorly constructed and the Figure 2c is meant to visualize the influence of withdrawal on selected metabolites and also show the effect of chronic alcohol intake on the selected metabolites at baseline. The description of the Figure 2c has been modified in result section from line 156 onwards: “Overall, Fig. 2c demonstrates that a number of identified metabolites altered in sAUD patients relative to control are affected by alcohol withdrawal. Apart from 4-pyridoxic acid, cotinine, and heme metabolites bilirubin and biliverdin, the shifts observed in the selected metabolites are generally in the opposite direction as compared to the baseline.”

The authors attempt to extend the significance of their findings by assessing post-mortem brain tissues from AUD subjects; however, the finding that many of the metabolites changed in T2/T1 are also present in AUD brain tissues is interesting; however, not strongly supporting of the authors' claims that these metabolites are markers of AUD (line 173). Concerning the plasma cohort itself, it is unclear how the authors assessed for compliance with alcohol withdrawal or whether the subjects' blood-alcohol levels were independently verified.

We did not claim that the metabolites significantly correlated with the psychological symptoms - and present in central nervous system (frontal cortex or CSF) - are “markers of AUD”. Line 173 did not refer to this idea, and the terms “markers of AUD” do not appear in the whole manuscript.

Regarding the compliance with alcohol cessation, we did not assess the ethanol blood level. The patients are hospitalized for a 3-week detoxification program, they are not allowed to drink alcohol and are under strict control of the nurses and medical staff of the unit. Consuming alcoholic beverage within the hospitalization unit is a reason for exclusion. However, we carefully monitored the liver function during alcohol withdrawal. For the reviewers’ information, we have added here below, the evolution of liver enzymes (ALT, AST, gGT) during the 3-week detoxification program as indirect markers of alcohol abstinence.

**Author response image 2. sa4fig2:** Data are described as median ± SEM. AST, Aspartate transaminase; ALT, Alanine transaminase; gGT: gamma glutamyltranspeptidase. ** p<0.01 vs T1, *** p<0.001 vs T1

The second area of concern is the need for more description of the analytical methodology, the lack of metabolite identification validation evidence, and related statistical questions. The authors cite reference #59 regarding the general methodology. However, this reference from their group is a tutorial/review/protocol-focused resource paper, and it is needs to be clarified how specific critical steps were actually applied to the current plasma study samples given the range of descriptions provided in the citations. The authors report a variety of interesting metabolites, including their primary fragment intensities, which are appreciated (Supplementary Table 3), but no MS2 matching scores are provided for level 2 or 3 hits. Further, level 1 hits under their definition are validated by an in-house standard, but no supporting data are provided besides this categorization. Finally, a common risk in such descriptive studies is finding spurious associations, especially considering many factors described in the current work. These include AUD, depression, anxiety, craving, withdrawal, etc. The authors describe the use of BH correction for multiple-hypothesis testing. However, this approach only accounts for the many possible metabolite association tests within each comparison (such as metabolites vs depression). It does not account for the multi-variate comparisons to the many behavior/clinical factors described above. The authors should employ one of several common strategies, such as linear mixed effects models, for these types of multi-variate assessments.

The methodological details related to the sample processing, data acquisition, data pre-processing and metabolite identification have been provided in the supplementary materials and described below. Supplementary table 3 has been amended with characteristic MS2 fragments for both positive and negative ionization modes if data was available. Additionally, all annotations against the in-house library additions have been rechecked, identification levels corrected and EICs for all level 1 identifications are provided in the supplementary material.

As described in the statistical analysis methods, BH correction was employed in the group-wise comparisons to shortlist the altered features for identification. Manual curating was then applied for the significant features and annotated metabolites subjected to correlation analysis. In this discovery-based approach the aim was to discover potential candidates linked with psychological symptoms for subsequent work to evaluate causality. Hence, the application of multi-variate analysis assessing biomarker candidates is not in the scope of this study.

*“LC-MS analysis.* Plasma sample preparation and LC-MS measurement followed the parameters previously detailed in Klåvus et al (57). Samples were randomized and thawed on ice before processing. 100 µl of plasma was added to 400 µl of LC-MS grade acetonitrile, mixed by pipetting four time, followed by centrifugation in 700 *g* for 5 minutes at 4 °C. A quality control sample was prepared by pooling 10 µl of each sample together. Extraction blanks having only cold acetonitrile and devoid of sample were prepared following the same procedure as sample extracts. LC-MS grade acetonitrile, methanol, water, formic acid and ammonium formate (Riedel-de Haën, Honeywell, Seelze, Germany) were used to prepare mobile phase eluents in reverse phase (Zorbax Eclipse XDBC18, 2.1 × 100 mm, 1.8 μm, Agilent Technologies, Palo Alto, CA, USA) and hydrophilic interaction (Acquity UPLC BEH Amide 1.7 μm, 2.1 × 100 mm, Waters Corporation, Milford, MA, USA) liquid chromatography separation. In reverse phase separation, the samples were analyzed by Vanquish Flex UHPLC system (Thermo Scientific, Bremen, Germany) coupled to high-resolution mass spectrometry (Q Exactive Focus, Thermo Scientific, Bremen, Germany) in both positive and negative polarity mass range from 120 to 1200, target AGC 1e6 and resolution 70,000 in full scan mode. Data dependent MS/MS data was acquired for both modes with target AGC 8e3 and resolution 17,500, precursor isolation window was 1.5 amu, normalized collision energies were set at 20, 30 and 40 eV and dynamic exclusion at 10.0 seconds. In hydrophobic interaction separation, the samples were analyzed by a 1290 LC system coupled to a 6540 UHD accurate mass Q-ToF spectrometer (Agilent Technologies, Waldbronn, Karlsruhe, Germany) using electrospray ionization (ESI, Jet Stream) in both positive and negative polarity with mass range from 50 to 1600 and scan rate of 1.67 Hz in full scan mode. Source settings were as in the protocol. Data dependent MS/MS data was acquired separately using 10, 20 and 40 eV collision energy in subsequent runs. Scan rate was set at 3.31 Hz, precursor isolation width of 1.3 amu and target counts/spectrum of 20,000, maximum of 4 precursor pre-cycle, precursor exclusion after 2 spectra and release after 15.0 seconds. Detectors were calibrated prior sequence and continuous mass axis calibration was performed throughout runs by monitoring reference ions from infusion solution for operating at high accuracy of < 2 ppm. Quality control samples were injected in the beginning of the analysis to equilibrate the system and after every 12 samples for quality assurance and drift correction in all modes. All data were acquired in centroid mode by either MassHunter Acquisition B.05.01 (Agilent Technologies) or in profile mode by Xcalibur 4.1 (Thermo Fisher Scientific) softwares.

Metabolomics analysis of TSDS frontal cortex and CSF samples using the same 1290 LC system coupled with a 6540 UHD accurate mass Q-ToF spectrometer has been previously accomplished by Karkkainen et al (10).

*Peak picking and data processing.* Raw instrumental data (*raw and *.d files) were converted to ABF format using Reifycs Abf Converter (https://www.reifycs.com/AbfConverter). MS-DIAL (Version 4.70) was employed for automated peak picking and alignment with the parameters according to Klåvus et al., 2020 (57) separately for each analytical mode. For the 6540 Q-ToF mass data minimum peak height was set at 8,000 and for the Q Exactive Focus mass data minimum peak height was set at 850,000. Commonly, *m/z* values up to 1600 and all retention times were considered, for aligning the peaks across samples retention time tolerance was 0.2 min and MS1 tolerance 0.015 Da and the “gap filling by compulsion” was selected. Alignment results across all modes and sample types as peak areas were exported into Microsoft Excel sheets to be used for further data pre-processing.

Pre-processing including drift correction and quality assessment was done using the notame package v.0.2.1 R software version 4.0.3 separately for each mode. Features present in less than 80% of the samples within all groups and with detection rate in less than 70% of the QC samples were flagged. All features were subjected to drift correction where the features were log-transformed and a regularized cubic spline regression line was fitted for each feature against the quality control samples. After drift correction, QC samples were removed and missing values in the non-flagged features were imputed using random forest imputation. Finally, the preprocessed data from each analytical mode was merged into a single data matrix.

Molecular feature characteristics (exact mass, retention time and MS/MS spectra) were compared against in-house standard library, publicly available databases such as METLIN, HMDB and LIPIDMAPS and published literature. Annotation of metabolites and the level of identification was based on the recommendations given by the Chemical Analysis Working Group (CAWG) Metabolomics Standards Initiative (MSI) (59): 1 = identified based on a reference standard, 2 = putatively annotated based on physicochemical properties or similarity with public spectral libraries, 3 = putatively annotated to a chemical class and 4 = unknown.”

Reference 59: Sumner LW, Amberg A, Barrett D, Beale MH, Beger R, Daykin CA, et al. Proposed minimum reporting standards for chemical analysis. Metabolomics. 2007;3:211–221.

**Recommendations for the authors:**

**Reviewer #1:**
(1) There should be more discussion comparing and contrasting the differences between the 2 cohorts (ALCOHOLBIS versus GUT2BRAIN), instead of stressing the similarities.

As indicated in the results section, we have verified that the ALCOHOLBIS cohort and GUT2BRAIN cohort are similar in term of age, gender, smoking habits, drinking habits and severity of psychological symptoms. Those similar features are important to allow the combination of the metabolomics data from the two cohorts, which subsequently allows to have a bigger sample size (n = 96) and more statistical power.

(2) The identification of 97 heavy alcohol users based on hospital codes at autopsy may not be the most rigorous way to define those with AUD. More information is needed on how these 97 were classified as heavy alcohol users.

The classification of subjects to the group who have a history of heavy alcohol use was not based solely on the autopsy records. The classification was also based on medical history, which in Finland is available from the whole life of the subjects, and including diagnoses and laboratory finding. The subjects needed to have a diagnosis of alcohol-related disease, as stated in the methods section of the manuscript. However, since some of the used diagnoses are related to organ damage related to heavy alcohol use, we do not claim that these subjects would all have alcohol dependence. But history of heavy use of alcohol is needed to get organ damage associated with alcohol use. Therefore, we consider that diagnosis of alcohol-related disease is a clear sign of a history of heavy alcohol use.

(3) The fact that the control group mainly died of cardiovascular disease confounds the interpretations around alcohol impact metabolite levels. How much of the metabolomics differences are related to hyperlipidemia or other CVD risk factors in the controls?

There are no healthy controls in post-mortem studies, since all subjects need to die from something to be included to the cohort. The challenge in studying AUD is that they die relatively young. The only other group of individuals who die outside of hospital at the relatively same age as subjects with AUD are those with CVD. Post-mortem autopsies are done in Finland to all who die outside of hospital, and these are the main source of samples for post-mortem sample cohorts. Therefore, there is no other control group to compare AUD subject to in these types of studies.

As for the altered metabolites in the post-mortem sample, the phospholipids observed could be associated with CVD. However, alterations in phospholipids are also commonly associated with alcohol use and AUD (for a review see (Voutilainen and Kärkkäinen, 2019)) and this effect is also seen in the results from the clinical cohorts in this study (Figure 1). Therefore, it cannot be said that these phospholipids finding would be due to selection of the control group.

(4) When examining metabolomics alterations, it is extremely important to understand what people are eating (i.e., providing a substrate). A major confounding issue here is that heavy alcohol users typically choose drinking over eating food. How much of the observed alterations in the plasma metabolome is due to the decreased food intake? Some validation in animal models of ethanol exposure compared to pair-fed controls would help strengthen causal relationships between metabolites and alterations in the circulation and CNS.

Regarding the validation in animal models of ethanol exposure, we were very careful in our discussion to avoid pretending that the study allowed to test causality of the factors. This was certainly not the objective of the present study. The testing of causality would indeed probably necessitate animal models but these models could only test the effects of one single metabolite at a time and could not at the same time capture the complexity of the changes occurring in AUD patients. The testing of metabolites would be a totally different topic. Hence, we do not feel comfortable in conducting rodent experiments for several reasons. First, AUD is a very complex pathology with physiological and psychological/psychiatric alterations that are obviously difficult to reproduce in animal models. Secondly, as mentioned by the reviewer, AUD pathology spontaneously leads to nutritional deficits, including significant reductions in carbohydrates, lipids, proteins and fiber intakes. We have recently published a paper in which we carefully conducted detailed dietary anamneses and described the changes in food habits in AUD patients (Amadieu et al., 2021). As explained below, some blood metabolites that are significantly correlated with depression, anxiety and craving belong to the xanthine family and are namely theobromine, theophylline, and paraxanthine, which derived from metabolism of coffee, tea or chocolate (which are not part of the normal diet of mice or rats).Therefore, conducting an experiment in animal model of ethanol exposure compared to pair-fed controls will omit the important impact of nutrition in blood metabolomics and consequently won’t mimic the human AUD pathology. In addition, if we take into consideration the European Directive 2010/63/EU (on the protection of animals used for scientific purposes) which aims at Reducing (Refining, Replacing) the number of animals used in experiment, it is extremely difficult to justify, at the ethical point of view, the need to reproduce human results in an animal model that won’t be able to mimic the nutritional, physiological and psychological alterations of alcohol use disorder.

As explained above, we do agree with the reviewer that AUD is not only “drinking alcohol” but is also associated with reduction in food intake that obviously influenced the metabolomics data presented in this current study. We have therefore added some data, which have not been published in the previous version of the manuscript, in the results section that refer to key nutrients modified by alcohol intake and we refer to those data and their link with metabolomics in the discussion section:

Results section page 8, Line 153-155. This sentence has been added:

“The changes in metabolites belonging to the xanthine family during alcohol withdrawal could be explained by the changes in dietary intake of coffee, tea and chocolate (see Fig S5).”

Discussion section: Page 11, Line 234-238.

“Interestingly, the caffeine metabolites belonging to the xanthine family such as paraxanthine, theophylline and theobromine that were decreased at baseline in AUD patients compared to controls, increased significantly during alcohol withdrawal to reach the levels of healthy controls. Changes in dietary intake of coffee, tea and chocolate during alcohol withdrawal could explain these results”.

In the conclusion, Page 16, Line 360-32, we clearly stated that: “LC-MS metabolomics plasma analysis allowed for the identification of metabolites that were clearly linked to alcohol consumption, and reflected changes in metabolism, alterations of nutritional status, and gut microbial dysbiosis associated with alcohol intake”

Reference:

Amadieu C, Leclercq S, Coste V, Thijssen V, Neyrinck AM, Bindels LB, Cani PD, Piessevaux H, Stärkel P, Timary P de, Delzenne NM. 2021. Dietary fiber deficiency as a component of malnutrition associated with psychological alterations in alcohol use disorder. *Clinical Nutrition*
**40**:2673–2682. doi:10.1016/j.clnu.2021.03.029

Leclercq S, Cani PD, Neyrinck AM, Stärkel P, Jamar F, Mikolajczak M, Delzenne NM, de Timary P. 2012. Role of intestinal permeability and inflammation in the biological and behavioral control of alcohol-dependent subjects. *Brain Behav Immun*
**26**:911–918. doi:10.1016/j.bbi.2012.04.001

Leclercq S, De Saeger C, Delzenne N, de Timary P, Stärkel P. 2014a. Role of inflammatory pathways, blood mononuclear cells, and gut-derived bacterial products in alcohol dependence. *Biol Psychiatry*
**76**:725–733. doi:10.1016/j.biopsych.2014.02.003

Leclercq S, Matamoros S, Cani PD, Neyrinck AM, Jamar F, Stärkel P, Windey K, Tremaroli V, Bäckhed F, Verbeke K, de Timary P, Delzenne NM. 2014b. Intestinal permeability, gut-bacterial dysbiosis, and behavioral markers of alcohol-dependence severity. *Proc Natl Acad Sci U S A*
**111**:E4485–E4493. doi:10.1073/pnas.1415174111

Voutilainen T, Kärkkäinen O. 2019. Changes in the Human Metabolome Associated With Alcohol Use: A Review. *Alcohol and Alcoholism*
**54**:225–234. doi:10.1093/alcalc/agz030

**Reviewer #2:**
(1) More methodological information about the laboratory processing of samples, instrumentation, and data analysis needs to be provided. Reference 59 needs to be more specific and include important methodological details for this project. Please provide an actual methods section for the mass-spectrometry-based metabolomics.

The reviewer is correct that the methods should be described in detail but due to word limits, the description was moved to a supplementary file. Methodological details are provided in the answer to the final comment in the public reviews section and we kindly refer to that for the methodological details. Reference 57 (Klåvus et al) is a method paper and covers the whole untargeted metabolomics pipeline that is used in our work.

(2) The VIP figures, e.g., Figure 1b and Figure 2b are not very informative and would be better represented in a supplementary table

VIP scores for all annotated metabolites are provided in the supplementary table 3 along with peak data and other values derived from statistical tests. Furthermore, we have removed the VIP value in figures 1 and 2 and we have replaced them by an updated Volcano plot to represent also the VIP values in addition to the q and Cohen’s d values.

(3) The findings on odd-chain lyso-lipids are interesting, and while these have been reported biologically, odd-chain lipids are uncommon and should be validated with authentic standards as available (please provide an XIC of the level 1 peak and standard if possible, e.g., LPC 17:0) or at least a supplementary figure on manual inspection of the negative mode MS2 spectrum showing the putative fatty acid chain fragment. The current assignments are based on positive mode lipid class fragments and accurate mass.

We thank the reviewer for pointing this out and it is correct that the negative MS2 spectrum is essential for lipid identification. Although the current assignments show only positive fragments for many lipids, the fatty acid chain, if reported, has been confirmed from negative mode MS2 spectrum. The supplementary table 3 with peak information has been augmented with fragment information from both negative and positive ionizations if available. Also, reference and experimental MS2 spectra have been provided as separate supplemental file for level 1 identifications, including the odd-chain lyso-lipids LPC 15:0 and 17:0.

(4) Please provide some supplementary information (MS1/MS2 if available) on the untargeted features of interest (up and down-regulated) from Figure 1C, especially the 5 encircled features. If any manual annotation of these features was attempted, please include a brief description in the results/discussion.

All statistically significant features with MS2 data have been subjected to manual annotation and database searches using at least METLIN, HMDB and LipidMaps. Additionally, if the manual inspection failed to provide any identification, *in silico* fragmentation software MS-FINDER was used to calculate candidate molecular formula. The features were labeled as unknown if all efforts were unsuccessful. The peak characteristics of the key unknowns in Figure 1b have also been included in the supplemental table.

A note of the manual inspection has been included in the result section line 129: “The top-ranked metabolites in Fig. 1b remained unknown regardless of manual curation.”

**Reviewer #3:**
I think this is an interesting paper with a very solid methodology and an abundance of results. I am not an expert on metabolomics, and I have some very interesting hours here, trying (but sometimes failing) to grasp this paper's content. This paper also needs to be closely read by a reviewer who knows the metabolomics field and can give feedback on the meaning of the results. I have focused purely on the AUD clinical side as this is where I may contribute. My main concern is conceptualizing the aims and what authors want to investigate. As far as I understand, this is a study of the relationship between alcohol use and the metabolome, and in this respect, I think there are some issues.Just take the abstract that talks about (in the first sentence) alcohol use disorder ("AUD") - a term that generally sometimes refers to harmful use of alcohol and alcohol addiction and sometimes to all F10-diagnosis (and thus an inaccurate term), then the following sentence talks about what leads to alcohol addiction (not dependence) - and this in a mechanistic direction and in the last part of the second sentence talks about metabolomics being able to decipher metabolic events related to AUD. So, even in the first two sentences, it is confusing - is this about correlates, mechanisms, prevention, or treatment? The inaccuracy of terms continues in sentence 4. We have "chronic alcohol abuse" (?) and "severe alcohol use disorder (AUD)" (abbreviated for the second time). Later, only "alcohol abuse" is used and the abstract ends with something about these findings being interesting in "the management of [...] AUD". All this illustrates that there is a large mixture of concepts - what aspect of alcohol use or abuse are you looking at? Moreover, of intention: is it to find correlates, explanations, or targets for interventions? Without clarity in this respect, one can get lost in what all these interesting measures mean - how we should interpret them. This comment is made only for the abstract. However, but it is equally valid and important for the introduction and discussion parts of the ms, where additional terms and formulations are introduced: "heavy alcohol use" (lines 86-7) and "prevent or treat psychiatric disorders such as AUD" (lines 90-1). This is then reflected in the discussion where the authors claim that what they have found is related to "chronic alcohol abuse" (line 188), "heavy alcohol drinkers" (line 191), and "AUD patients" (lines 199 and 202 and further on).

We thank the reviewer for this useful comment and we apologize for the confusion. We agree that it is important to use the correct terms and definitions. All patients included in this study were diagnosed as severe AUD (for more information on the diagnosis, see answer to the comments related to DSM-IV and DSM5). This manuscript is consequently related to severe AUD and other terms like “alcohol abuse, “alcohol addiction” are therefore not appropriate. In the revised version of the manuscript, we have used severe AUD or the abbreviation sAUD. The figure and legends have been changed accordingly.

In the first paragraph of the results section, ALCOHOLBIS and GUT2BRAIN are compared. It says they are similar on many measures, including craving, but different on some measures, again including craving. It is difficult to grasp this even if the authors try to explain (lines 101-2). This sentence also introduces some discussion in the results section by saying something normative about their finding and relating this to other research (references 12, 13, and 14).

We would like to apologize for the confusion related to first paragraph of the results section. We have indeed indicated that, while the ALCOHOLBIS cohort and the GUT2BRAIN cohort are highly similar in term of biological and psychological features, a significant difference does exist in the compulsive component of the craving score. Indeed, the mean score of compulsion is 11 ± 3 in the ALCOHOLBIS cohort and 14 ± 3 in the GUT2BRAIN cohort. In healthy controls, the mean score of compulsion is 1.5 ± 1.5. Despite the statistically significant difference in craving between both cohorts, we do not think that this difference is relevant in our context since both scores (11 and 14) are considered high compared to the control group. In order to simplify the message, we have revised the first paragraph as follows:

“Both groups of patients were similar in terms of age, gender, smoking and drinking habits and presented with high scores of depression, anxiety and alcohol craving at T1 (Table 1). These biological and psychological similarities allow us to combine both cohorts (and consequently increase sample size) and compare them to a group of heathy controls for metabolomics analysis”.

In line 104 the abbreviation PCA is introduced but needs to be explained. Such objections could be made for many of the abbreviations used (sPLS-DA VIP, LPC, CSF, CNS, LPE, etc.), but of course, they may be made more difficult by the unusual way of stacking the different sections.

We thank the reviewer for pointing these out. Most abbreviations are written out in the figure legends or method section but indeed the organization of the different sections makes it less evident. The abbreviations pointed out have been opened in the results section when they are first used.

Furthermore, they say that the severity of AUD was "evaluated by a psychiatrist using the Diagnostic and Statistical Manual of Mental Disorders (DSM) criteria, fourth edition (DSM-IV) (ALCOHOLBIS cohort) or fifth edition (DSM-5)" (GUT2BRAIN cohort): This makes sense for DSM-5 but needs to be explained more for DSM-IV. They also need to say what levels were included.

We thank the reviewer for this very appropriate remark that deserves some explanations.

While the patients of the GUT2BRAIN cohort were enrolled in 2018-2019 where the DSM5 was applicable, the patients from the ALCOHOLBIS cohort were recruited many years before. The protocol related to the ALCOHOLBIS cohort was written before 2013, and approved by ethical committee, where the DSM-IV was the last version of the DSM used at that moment.

We therefore totally agree with the reviewer that our sentence “the severity of AUD was "evaluated by a psychiatrist using the Diagnostic and Statistical Manual of Mental Disorders (DSM) criteria, fourth edition (DSM-IV) (ALCOHOLBIS cohort) or fifth edition (DSM-5)" (GUT2BRAIN cohort)” is not correct. Indeed, DSM-IV (before 2013) described two distinct disorders, alcohol abuse and alcohol dependence, while the DSM-5 integrates the two DSM-IV disorders into a single disorder called alcohol use disorder with mild (2 or 3 symptoms), moderate (4 or 5 symptoms) and severe (6 or more symptoms) sub-classifications.

In this present study, we have enrolled patients that received the diagnosis of alcohol dependence (DSM-IV criteria) or severe alcohol use disorder (DSM5 criteria).

We have changed the paragraph related to this issue into this new one:

“The severity of AUD was evaluated by a psychiatrist using the Diagnostic and Statistical Manual of Mental Disorders (DSM) criteria, fourth edition (DSM-IV) (Alcoholbis cohort) or fifth edition (DSM-5) (GUT2BRAIN cohort). Patients evaluated with the DSM-IV received the diagnosis of “alcohol dependence”, while the patients evaluated with the DSM-5 received the diagnosis of “severe alcohol use disorder” (6 or more criteria). To simplify, we used the term “sAUD” (for severe alcohol use disorder) that includes both diagnosis (sAUD and alcohol dependence)”.

I am unsure about the shared first co-authorship and the shared last co-authorship request, but I leave this up to the editors and the journal policies. Also, the order of the different parts may be correct (the M+M placed last) but is unusual for many journals. This is also up to the journal to decide.

As mentioned in the guidelines to authors, the method section should be included at the end of the manuscript.